# Deep learning-driven fragment ion series classification enables highly precise and sensitive de novo peptide sequencing

Daniela Klaproth-Andrade[1,2], Johannes Hingerl [1], Yanik Bruns[1], Nicholas H. Smith[1], Jakob Träuble[1], Mathias Wilhelm [2,3] & Julien Gagneur [1,2,4,5]

Unlike for DNA and RNA, accurate and high-throughput sequencing methods for proteins are lacking, hindering the utility of proteomics in applications where the sequences are unknown including variant calling, neoepitope identification, and metaproteomics. We introduce Spectralis, a de novo peptide sequencing method for tandem mass spectrometry. Spectralis leverages several innovations including a convolutional neural network layer connecting peaks in spectra spaced by amino acid masses, proposing fragment ion series classification as a pivotal task for de novo peptide sequencing, and a peptide-spectrum confidence score. On spectra for which database search provided a ground truth, Spectralis surpassed 40% sensitivity at 90% precision, nearly doubling state-of-the-art sensitivity. Application to unidentified spectra confirmed its superiority and showcased its applicability to variant calling. Altogether, these algorithmic innovations and the substantial sensitivity increase in the high-precision range constitute an important step toward broadly applicable peptide sequencing.

Liquid chromatography tandem mass spectrometry is the method of choice for identifying proteins at high throughput[1]. To this end, proteins are first digested into peptides whose mass-to-charge (*m/z*) ratios are determined in a first mass spectrum. Next, selected peptides are fragmented along their backbone bonds to generate series of peptide fragments whose *m/z* ratios can be identified in a second mass spectrum[2]. In principle, this spectrum allows the reconstruction of the peptide sequence by reading out the *m/z* differences between consecutive peaks of the same ion series[3,4]. In practice, this task is very hard due to missing peaks, contamination peaks, and because the ion series of the peaks are not known a priori. Peptide identification is greatly facilitated when the experimental spectrum is compared to expected spectra from a limited set of possible peptides, typically the in-silico digested proteome of an organism under study[5]. This strategy, which

requires a precomputed database of possible peptides, is called database search[6–8]. The vast majority of proteomics studies rely on database search, even though, by design, database search does not allow the identification of novel or unexpected peptides. This prevents proteomics from being efficiently used in applications where the peptide sequences are not known a priori. This concerns neoepitope identification[9], antibody sequencing[10], pathogen surveillance[11], microbial community studies[12], and paleontology[13]. Therefore, efficient de novo peptide sequencing algorithms, which aim to identify peptides directly from spectra without relying on any database, are highly needed.

Most de novo peptide sequencing algorithms implement a combinatorial optimization approach in which the peptide that best fits the spectra is searched for. Various peptide-spectrum match (PSM) scores, i.e. scores that assess how well a candidate peptide corresponds to a

[1]Computational Molecular Medicine, School of Computation, Information and Technology, Technical University of Munich, Garching, Germany. [2]Munich Data Science Institute, Technical University of Munich, Garching, Germany. [3]Computational Mass Spectrometry, School of Life Sciences, Technical University of Munich, Freising, Germany. [4]Institute of Human Genetics, School of Medicine, Technical University of Munich, Munich, Germany. [5]Computational Health Center, Helmholtz Center Munich, Neuherberg, Germany. e-mail: mathias.wilhelm@tum.de; gagneur@in.tum.de

given spectrum, combined with combinatorial optimization techniques including dynamic programming[14–18] and genetic algorithms[19,20] have been used to identify best-fitting peptides. Nevertheless, missing and contamination peaks have strongly limited the accuracy of those algorithms. Parallel to this work, we and others have leveraged deep learning to make major progress on the forward problem, i.e., predicting a spectrum given a peptide sequence[21–23]. While these algorithms do not predict contamination peaks, they can predict the peak intensities and missing peaks of a given peptide. Hence, their predictions can be leveraged to develop more discriminative PSM scoring functions for de novo peptide sequencing algorithms as in the algorithm pNovo3[24]. Complementary to these algorithms based on combinatorial optimization, neural networks that directly predict the sequence of a peptide from the spectrum have recently been proposed. This includes DeepNovo[25], PointNovo[26], and Casanovo[27]. However, despite these efforts, the performance of existing de novo peptide sequencing methods remains limited, notably with a poor sensitivity in the high-precision range[27]. Further methodological improvements are needed to increase the number of highly confident peptide sequence identifications in tandem mass spectrometry experiments.

Here, we present Spectralis, a method that combines several algorithmic innovations for de novo peptide sequencing. Spectralis builds on established concepts in the field, such as spectrum graphs[3] and PSM scoring functions based on fragmentation patterns[15,16,18], and leverages our deep learning model for spectrum prediction Prosit[22]. Thereby, Spectralis substantially increases recall at very high precision compared to state-of-the-art and thus makes de novo peptide sequencing more amenable for routine application.

## Results
### Overview of spectralis
At its core, Spectralis consists of a supervised learning task that we call bin reclassification (Fig. 1a). We reasoned that if the complete set of $m/z$ values of either the singly charged b-ion series or y-ion series were known, including at positions where no peak is present in the

spectrum, the peptide sequence could be recovered by reading out the $m/z$ differences of either series. In practice, Spectralis operates on discrete bins of 1 Dalton (Da), i.e. at the mass resolution of one proton. We denote the task of predicting whether one such 1-Da bin contains a peak of a particular ion series as bin classification. To make this supervised learning problem efficiently amenable to neural networks, we introduce the amino acid-gapped (AA-gapped) convolutional layer, in which filters have gaps corresponding to the mass of amino acids. The advantage is that a single AA-gapped convolutional layer connects bins of potentially successive ions of one singly charged ion series, which can be separated by as much as 186 1-Da bins (tryptophan mass). In contrast to the idea of stacking inputs shifted by the amount of mass[9], our AA-gapped convolutions connect positions spaced by amino acid masses throughout all layers and not only for the input layer. Bin classification can leverage existing de novo peptide sequencing methods by encoding their candidate peptides in the input and learning how to mend their incorrect bin classes, termed bin reclassification. While our bin reclassification models still leave too many ambiguities in an entire spectrum to reliably yield the full correct peptide, they turned out to be instrumental: First, we showed that incorporating the bin class predictions yielded an improved PSM score, which we named Spectralis-score. We demonstrated that applying Spectralis-score to rank peptides predicted by existing methods boosts recall at high precision. Second, we showed that new, more likely to-be correct candidate peptides can be constructed by generating peptide sequences that connect the most probable bins of an ion series. An evolutionary algorithm, Spectralis-EA, combines these two ideas, where Spectralis-score serves as fitness function and mutated peptides are obtained by drawing paths among the most likely bins (guided mutations, Fig. 1b).

We trained and evaluated the proposed methods using a dataset consisting of 7,902,759 spectra from 302,054 peptides identified with MaxQuant[28] at 1% false discovery rate (FDR) in 30 different healthy human samples[29]. The dataset was split into a train, validation, and test set such that no correct peptide and no experimental spectrum is shared between sets. Moreover, all methods were trained and

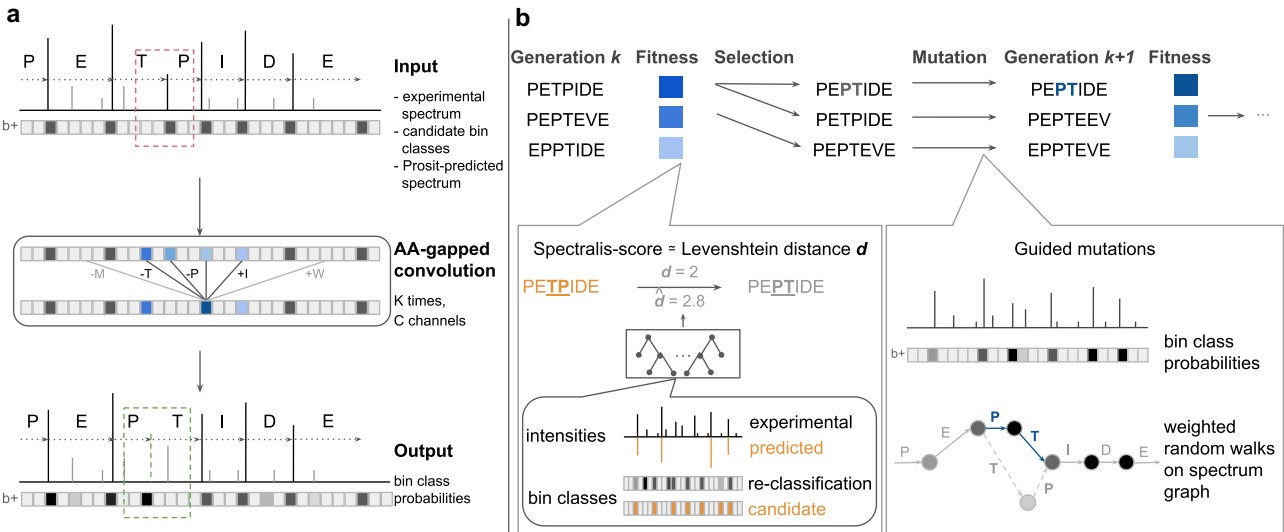

**Fig. 1 | Bin reclassification and overview of Spectralis. a** The deep learning architecture for bin reclassification consisting of AA-gapped convolutions for correcting erroneous bin classes (red box) of an input candidate peptide. Input for the model are the binned experimental intensities, the initial bin class labels for y-ions and b-ions, and the binned Prosit-predicted intensities for the input peptide sequence. The model outputs the probabilities for each bin to contain a peak labeled as a y-ion and as a b-ion. **b** Spectralis-EA is an evolutionary algorithm. Peptide sequences from a generation $k$ are selected

based on their fitness to define the next generation $k + 1$. The fitness, or Spectralis-score, is an estimate of the Levenshtein distance from the input peptide (orange) to the correct peptide (gray). It is obtained with a random forest taking features computed from the experimental and Prosit-predicted spectra and from the output of the bin reclassification model (left inset) as input. The peptides selected for the next generation are mutated by performing random walks along the spectrum graph favoring nodes stemming from bins with high probabilities (right inset).

**Article** https://doi.org/10.1038/s41467-023-44323-7

evaluated on the same splits (Methods). For the sake of consistency and clarity, we often present results for the heart sample, since the performance on this sample is similar to the median performance across samples. Furthermore, we provide performance plots of the best-performing and worst-performing samples in the Supplementary Material. We benchmarked our methods against the tool Novor[16], which outperforms the widely used commercial software PEAKS[17], and the recent method Casanovo[27].

## Bin reclassification improves ion series labeling

We first evaluated the performance of the bin reclassification model by comparing the initial bin class labels of singly charged b-ions and y-ions proposed by Novor and Casanovo to the bin classes predicted by our model. Experimental peak intensities, bin classes of the initial peptide, and peak intensities of the candidate peptide predicted by Prosit[22] served as input to the model. Figure 2a shows an example of a successful bin reclassification of a peptide sequence proposed by Casanovo: Our model correctly predicted all four bin changes that were needed to successfully transform the incorrect sequence into the correct one with a probability above 0.5.

Overall, our model achieved an area under the precision-recall curve of 0.69 for b-ions and 0.82 for y-ions after reclassifying Casanovo's initial bin classes of peptide sequences (Fig. 2b). The bin class labels proposed by our model improved upon the initial precision and recall given by Casanovo's initial bin classes for both b-ions and y-ions. At Casanovo's initial recall, the precision improved by 35% (from 0.62 to 0.84) for b-ions and by 25% (from 0.72 to 0.90) for y-ions. Moreover, at Casanovo's initial precision, the recall improved by 8% (from 0.62 to 0.67) for b-ions and by 14% (from 0.72 to 0.82) for y-ions. This indicated that the bin reclassification model could be used to correct predictions of Casanovo. To evaluate the performance of suggesting changes of the initial bin labels, we next evaluated the model with change precision-recall curves (Methods), which quantify model performance at accurately identifying bins that are initially incorrectly labeled and for which our model needs to predict a change. On the heart sample, we obtained an area under the change precision-recall curve of 0.57 for b-ions and 0.62 for y-ions. For y-ions, our model achieved a change recall of 0.66 at a change precision of 0.5 (Fig. 2c). At a change recall of 0.5, the model achieved a change precision of 0.78. Our model also improved bin class predictions of peptides proposed by Novor (Supplementary Fig. 1).

Similar improvements held across all samples for both Novor and Casanovo (Fig. 2d, Supplementary Fig. 2). We observed a median relative improvement in recall of 8% and 11% and precision of 16% and 28% for Novor and Casanovo, respectively (Fig. 2e). Our model achieved higher improvement in precision and recall for peptide sequences proposed by Casanovo in comparison to the ones proposed by Novor, even though Casanovo outperforms Novor when considering their performance at peptide level. We attribute this to the fact that incorrect sequences by Casanovo are generally longer than incorrect sequences by Novor, leading to a higher amount of incorrect initial bin classes, which our model is able to correct.

We investigated alternative models for bin reclassification based on convolutional neural networks with and without regular dilations, as well as hybrid approaches combining AA-gapped convolutions with regular convolutional layers and found that these underperformed the final approach consisting solely of AA-gapped convolutions (Supplementary Fig. 3). Altogether, these results showed that the final model for bin reclassification using AA-gapped convolutions could help correct initially incorrect candidate peptides.

## Bin reclassification allows generating improved candidate peptides

We leverage the bin reclassification model to modify peptides into additional, more promising, candidate peptides. To this end, our algorithm considers all bins predicted with a probability greater than a certain cutoff to contain a singly charged b-ion or y-ion. Next, it constructs a graph that connects highly probable bins that are spaced by a single amino acid mass. We observed that determining the peptide sequence from the path with the highest cumulated bin probabilities did not lead to an improvement in peptide recall (Supplementary Fig. 4). Therefore, we considered generating multiple additional peptides by performing weighted random walks on the graph, weighting more likely bins higher (Methods). The optimal bin probability threshold of 0.35 was determined by hyper-parameter search (Methods, Supplementary Fig. 5)

We used candidates from Casanovo as initial peptides and replaced them with candidates from Novor when the Casanovo peptide mass did not match the precursor mass, similar to the Casanovo author's suggestion[27]. Figure 3a shows an example in which, starting from an incorrect candidate peptide, the correct peptide was the most frequently generated peptide, comprising 32% of all 1024 generated candidates. Generally, the minimal Levenshtein distance among 1024 generated candidates was smaller than the Levenshtein distance of the corresponding initial peptide (Fig. 3b). Moreover, starting from candidate peptides with Levenshtein distances between 2 and 7, each random walk generated the correct peptide with a probability between 1% and 10% in median (Fig. 3c). The procedure was not able to correct sequences with a Levenshtein distance larger than 13 to the correct peptide sequence. However, considering the mean peptide length of 19.32 in the dataset, this would have corresponded to a substitution of a large part of the sequence.

Our procedure generated the correct peptide sequence for approximately half of the initial sequences when the initial sequences had a Levenshtein distance of two to the correct peptide sequence (Fig. 3d). This percentage decreased for initial sequences with larger Levenshtein distances. Collectively, these results indicate that this generation process could be used to iteratively improve candidate peptides, as well as to generate a population containing peptides closer to, when not yet equal to, the correct peptide in a single shot. We called this procedure, which allows evolving a candidate peptide into an improved one, guided mutation.

## Levenshtein distance estimate improves PSM scoring

Having established a promising algorithm that generates additional candidate peptides based on bin reclassification, we additionally considered using bin reclassification to estimate the correctness of any PSM, either generated by our algorithm or by existing de novo peptide sequencing tools. We and others have previously shown that rescoring PSMs using additional information, e.g., fragment intensity-based scores can be used effectively to separate incorrect from correct matches[21,22,30–32]. However, rather than estimating the confidence of a PSM (i.e. a peptide sequence mapping correct or incorrect to a spectrum), we here considered estimating the Levenshtein distance, i.e., the minimal number of elementary sequence edits separating two sequences[33], of a candidate peptide to the correct peptide of a given spectrum. Here and elsewhere, we considered a candidate peptide to be correct if it exactly matched the peptide identified by MaxQuant at 1% FDR up to isoleucine-leucine substitution. We reasoned that the Levenshtein estimates would yield a quantitative notion of how far a peptide sequence is from the correct one in sequence space, making it a valuable loss function for combinatorial optimization algorithms.

We trained a random forest to predict the Levenshtein distance of a candidate peptide to the correct peptide given 114 features including the number of predicted bin class changes by the bin reclassification model and the similarity between experimental and Prosit-predicted spectra (Methods, Supplementary Table 1). Overall, the method provided good estimates of the Levenshtein distance for peptides predicted by both Casanovo and Novor (Fig. 4a). No improved performance was achieved when fitting an XGBoost[34] model instead

Nature Communications | (2024)15:151 3

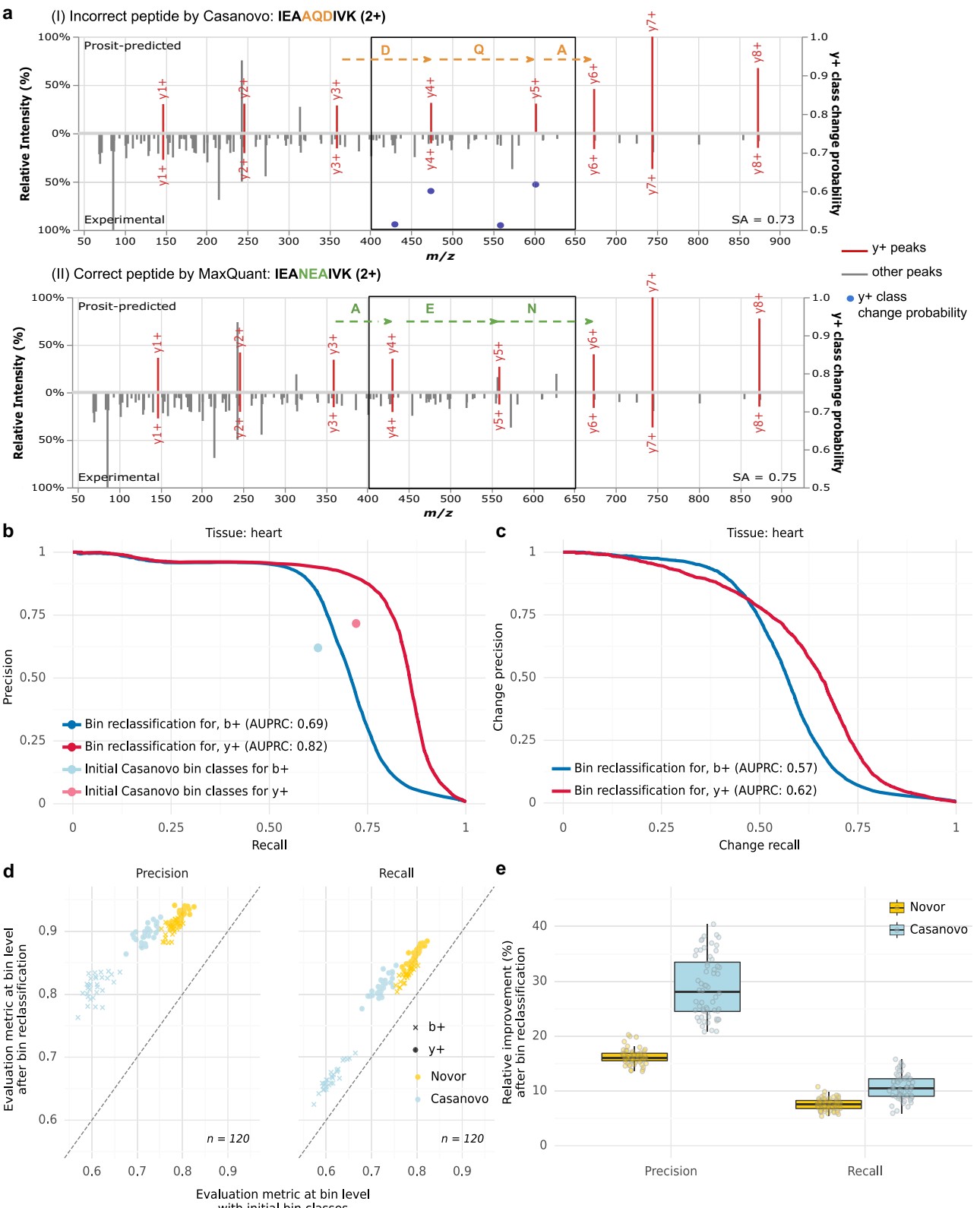

(Supplementary Fig. 6). We, therefore, continued with the random forest. A notable deviation was seen for sequences with a Levenshtein distance of 1, which typically had a mass differing from the mass of the correct peptide by more than 20 ppm, possibly because our training dataset did not include these instances and because the computed spectral angles, Pearson and Spearman correlation coefficients were generally lower than the ones for sequences with close Levenshtein distances (Supplementary Fig. 7). Importantly, the predicted distance for the correct peptides was smaller than 1 for the vast majority of correct peptides, indicating that our scoring function is able to separate correct from incorrect peptide sequences. Consistent with this observation, the estimated Levenshtein distance was smaller for the correct peptide than for corresponding incorrect candidates on around 93% of the evaluated spectra for both Novor and Casanovo

**Fig. 2 | Bin reclassification performance. a** An example of a bin reclassification: (I) Prosit-predicted spectrum (top) and experimental spectrum (bottom) for an incorrect peptide sequence IEA**AQD**IVK proposed by Casanovo. Singly charged y-ions (y +) are colored in red. Bin class change probabilities for y-ions are marked with blue dots (secondary *y*-axis, cropped at 0.5 to not clutter the plot). (II) Prosit-predicted spectrum (top) and experimental spectrum (bottom) for the correct peptide sequence IEA**NEA**IVK identified by MaxQuant at 1% FDR. The region where ion series label predictions differ between (I) and (II) is delimited with boxes. Incorrect residues are marked in orange. The residues differing in the correct sequence are marked in green. The spectral angle (SA) between the experimental and Prosit-predicted spectrum is indicated for both peptide sequences. **b** Precision-recall curves for bin reclassification of b-ions and y-ions after relabeling initial bin

classes proposed by Casanovo on the test set of the heart sample compared to the precision and recall computed at bin level for the initial bin class labeling. The average precision-recall is denoted as AUPRC. **c** As in (**b**) for change-precision-recall curves. **d** Precision (left) and recall (right) after bin reclassification against before when using Novor (yellow) or Casanovo (blue) for the initial peptide. Data on test sets across all *n* = 30 samples (different tissues). **e** Distribution of relative improvement of precision and recall after bin reclassification on the test sets of all *n* = 30 samples (different tissues) over Novor and Casanovo for b-ions and y-ions. The data in (**e**) are represented as boxplots in which the middle line indicates the median, the bounds of the box indicate the first and third quartiles and the whiskers indicate ±1.5 × IQR (interquartile range) from the third and first quartile, respectively. Source data are provided as a Source Data file.

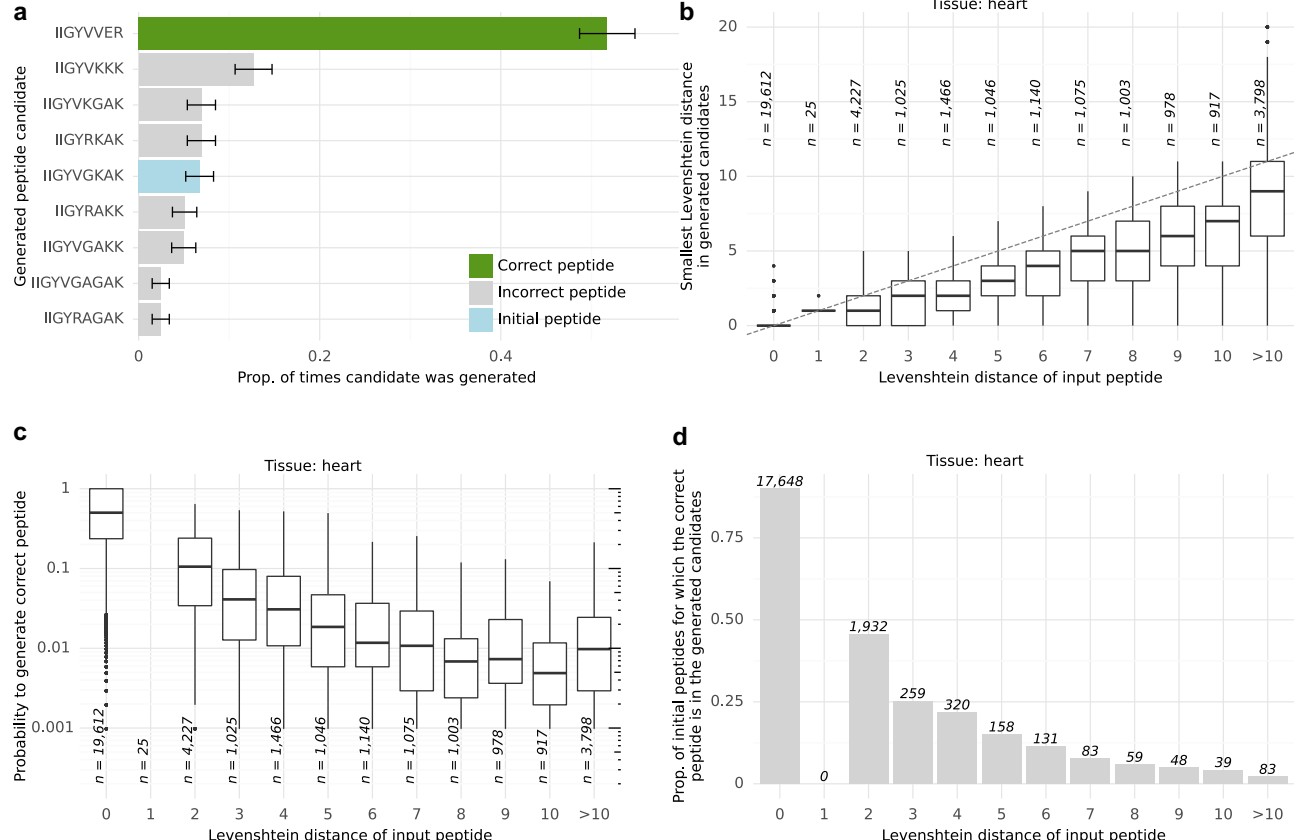

**Fig. 3 | Guided mutation performance. a** Proportion of times that a peptide sequence is generated based on the initial sequence IIGYV**GKAK** proposed by Casanovo out of *n* = 1024 generated candidate peptides with guided mutations. The green bar shows the proportion of times that the correct sequence IIGYV**VER** was generated. The blue bar shows the proportion of times that the initial sequence was generated. Error bars represent 95% confidence intervals of the performed two-sided binomial test. **b** Distribution of the smallest Levenshtein distance among 1024 guided mutations as a function of the Levenshtein distance of the initial peptides on the heart sample. **c** Distribution of the probability to generate the correct

peptide sequence among 1024 draws against Levenshtein distances of the initial peptide sequences on the heart sample. **d** Proportion of initial peptides for which the correct peptide sequence is generated at least once among 1024 draws as a function of the Levenshtein distances of the initial peptide sequences. The data in (**b**) and (**c**) are represented as boxplots in which the middle line indicates the median, the bounds of the box indicate the first and third quartiles and the whiskers indicate ±1.5 × interquartile range from the third and first quartile. Outlying data points are shown as dots. Source data are provided as a Source Data file.

(Fig. 4b). The good discriminability indicated that our Levenshtein distance prediction could be used as a cost function in a combinatorial optimization-based de novo peptide sequencing algorithm.

We named our Levenshtein distance estimator Spectralis-score. Rescoring PSMs using Spectralis-score instead of original scores by Novor and Casanovo consistently improved recall at all precisions, notably at high-precision ranges. For instance, the peptide recall at 90% precision increased by 76% (from 0.25 to 0.44) when compared to the initial ranking by Casanovo on the heart sample (Fig. 4c). Notably, the relative performance improvements in recall at 90% precision were

larger for longer peptides, reflecting the increase difficulties of de novo tools to perfectly sequence long peptides (Supplementary Fig. 8). Moreover, Spectralis-score outperformed the ranking given by the spectral angles between experimental and Prosit-predicted spectra, which is a feature of the model, or using PredFull-predicted[35] spectra (Supplementary Fig. 9). Further integrating the spectral angles with PredFull-predicted spectra into a combined score did not lead to any improvement (Supplementary Fig. 9, Methods). This shows that the bin reclassification model provides complementary information to mere spectrum predictions. Since the mass of the peptides proposed

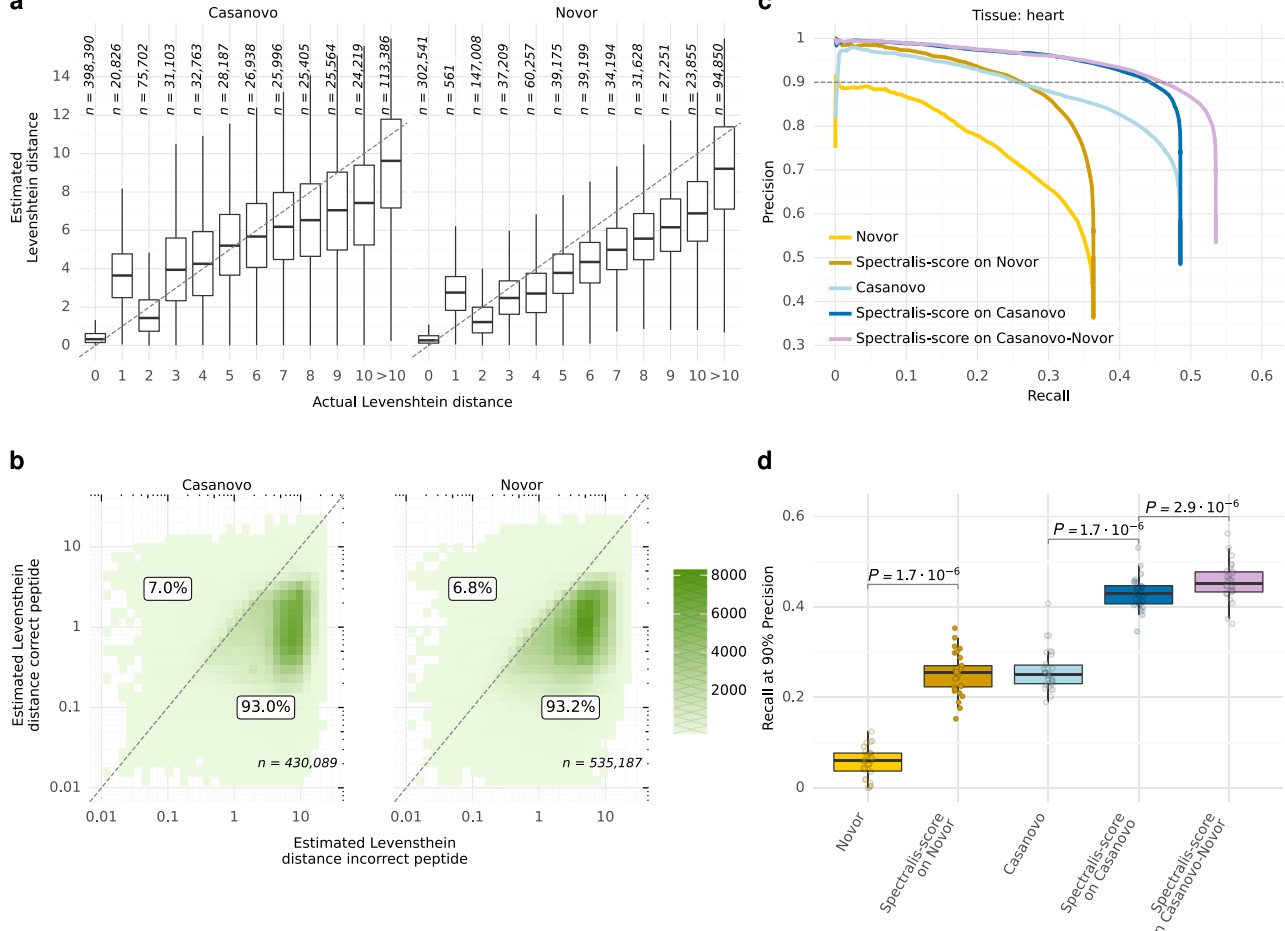

**Fig. 4 | Levenshtein distance estimator performance. a** Estimated against actual Levenshtein distances of incorrect peptide identifications by Novor and Casanovo to the correct peptide sequence by MaxQuant across all 30 human samples. **b** Estimated Levenshtein distances of incorrect peptides against the estimated Levenshtein distances of the corresponding correct peptide sequences for a given spectrum across all 30 samples. The percentage of points above or on the diagonal line and below the diagonal line is labeled. **c** Precision-recall curves at peptide level before and after rescoring peptide identifications by Novor and Casanovo on the heart sample with Spectralis-score, our Levenshtein distance estimator, including

the precision and recall for peptides from Casanovo with Novor substitutes for peptides with wrong mass (Casanovo-Novor). **d** Recall at 90% precision before and after rescoring peptides from Novor, Casanovo, as well as the combination of Casanovo and Novor sequences (Casanovo-Novor) across all $n = 30$ samples (different tissues). Statistical significance from a two-sided paired Wilcoxon test. The data in (**a**) and (**d**) are represented as boxplots in which the middle line indicates the median, the bounds of the box indicate the first and third quartiles and the whiskers indicate $\pm 1.5 \times$ IQR (interquartile range) from the third and first quartile, respectively. Source data are provided as a Source Data file.

by Casanovo did not match the precursor mass in approximately half of the spectra (44% in the heart sample), we replaced these peptides with Novor peptides, similarly to the suggestion by the authors of Casanovo, and denoted this combination of peptides Casanovo-Novor. We applied Spectralis-score to this combination and achieved higher recall at all precision ranges (Fig. 4c). A similar performance increase was observed for all other human samples (Fig. 4d, Supplementary Fig. 10).

The performance improvement of Spectralis-score held consistently after stratifying by precursor charge state and peptide length (Supplementary Figs. 11, 12). Furthermore, applying Spectralis-score to peptides proposed by the de novo sequencing tools DeepNovo[25] and PointNovo[26] on the heart sample also improved recall at all precision ranges when compared to the original scores at peptide level (Supplementary Fig. 13).

Finally, we assessed the generalization of Spectralis-score trained on human peptides to a dataset spanning nine different species and different experimental configurations (Methods)[25]. Without retraining, Spectralis-score improved the average precision (AUPRC) on all but one species and the recall at 90% precision on seven of nine species (Supplementary Figs. 14, 16–17). Rescoring peptides identified by

PointNovo improved the AUPRC on all but one species (Supplementary Figs. 15–17) and the recall at 90% precision on seven of nine species. The improvements have modest amplitude. Nonetheless, these results indicate the robustness of Spectralis-score as it was neither trained using DeepNovo and PointNovo candidate peptides nor on the nine-species dataset.

## An evolutionary algorithm increases the sensitivity of de novo peptide sequencing

Next, we designed an evolutionary algorithm, Spectralis-EA, that integrates guided mutations and Spectralis-score. For a given spectrum, the evolutionary algorithm starts from an initial population of candidate peptides derived from Casanovo-Novor candidates. At each iteration, a subset of the peptides is randomly selected for the next generation, whereby the peptides with smaller predicted Levenshtein distances are more likely to be selected. Moreover, selected peptides are mutated using the guided mutation procedure. The evolutionary algorithm is run for five generations and the best-scoring peptide is reported. The optimal population size was determined by hyperparameter search (Methods, Supplementary Fig. 5). The performance of the Levenshtein distance estimator remained stable across

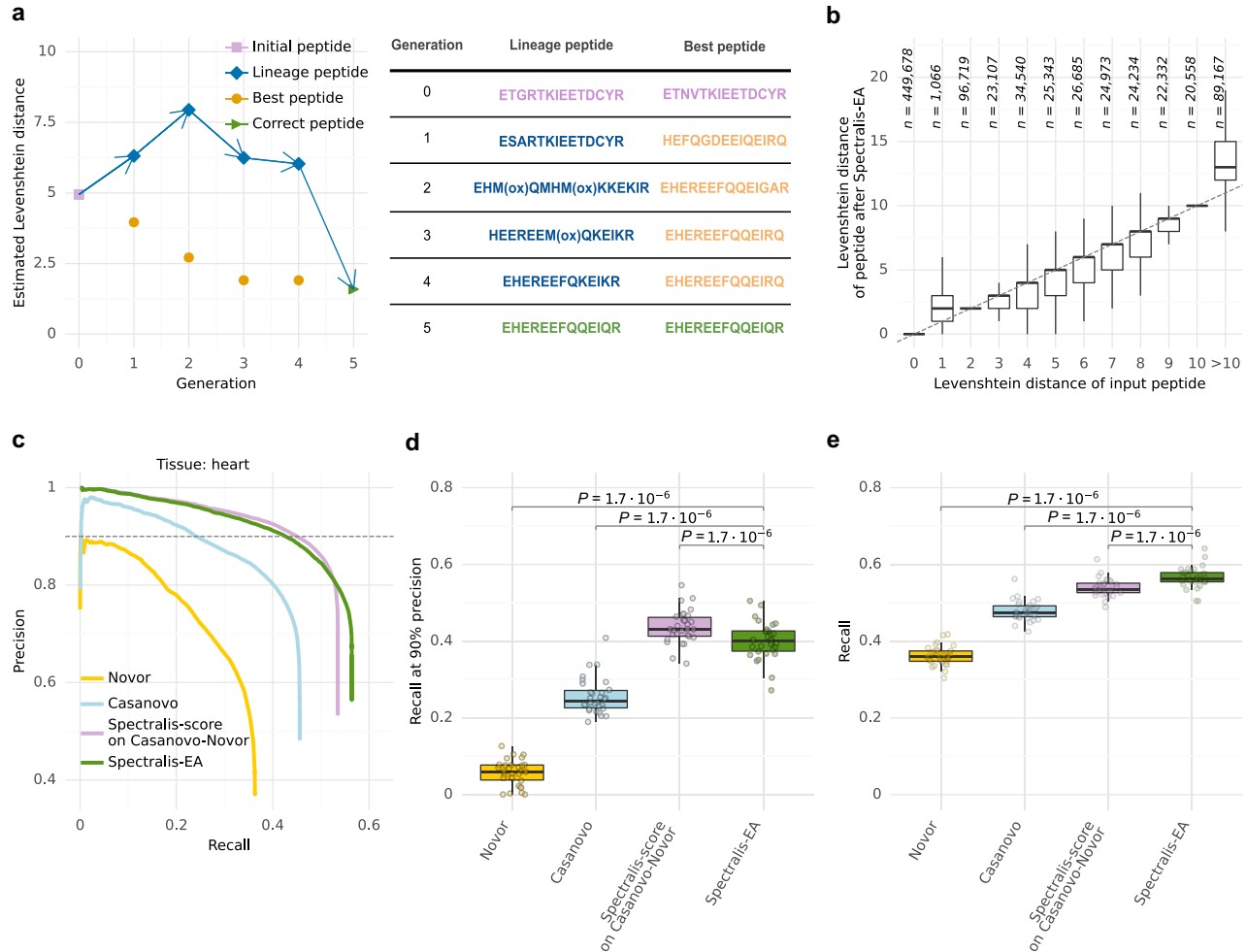

**Fig. 5 | Spectralis-EA performance. a** Example of a successful optimization of the initially incorrect peptide sequence ETGRTKIEETDCYR predicted by Novor after 5 generations of the evolutionary algorithm. For each generation, the estimated Levenshtein distances are provided for the best peptide (i.e., most highly scored peptide), and the lineage peptide (i.e. the candidate peptide leading to the correct peptide sequence). **b** Levenshtein distances of input peptide sequences against Levenshtein distances of peptide sequences returned by Spectralis-EA. **c** Precision-recall curves of identifications at peptide level for Novor, Casanovo, and Spectralis-EA, as well as Spectralis-score on the combination of Casanovo and Novor sequences (Casanovo-Novor) on the test set of the heart sample. **d** Recall at 90%

precision for Novor, Casanovo, Spectralis-score on the combination of Casanovo and Novor (Casanovo-Novor) and Spectralis-EA on the test sets of all 30 samples. **e** Overall recall for Novor, Casanovo, Spectralis-score on Casanovo-Novor, and Spectralis-EA on the test sets of all $n = 30$ samples (different tissues). Statistical significance for (**d**) and (**e**) from a two-sided paired Wilcoxon test. The data in (**b**), (**d**) and (**e**) are represented as boxplots in which the middle line indicates the median, the bounds of the box indicate the first and third quartiles and the whiskers indicate ±1.5 × IQR (interquartile range) from the third and first quartile, respectively. Source data are provided as a Source Data file.

generations (Supplementary Fig. 18). This was important as it could otherwise mislead the optimization algorithm. Investigations showed that the evolutionary algorithm could not optimize initial peptides with large predicted Levenshtein distances. Furthermore, it tended to generate incorrect peptides with better scores than initially correct peptides for candidate peptides with small estimated Levenshtein distances. Therefore, we opted to apply the evolutionary algorithm only to initial peptides with an estimated Levenshtein distance between 1 and 7 and to report initial peptides otherwise.

Figure 5a shows a successful example of optimizing an initially incorrect peptide sequence. Remarkably, when applied to the heart sample, Spectralis-EA improved candidate peptides, i.e., returned a candidate peptide with a smaller Levenshtein distance, for 31.6% of initially incorrect peptides with Levenshtein distances smaller than 10 (Fig. 5b). However, the evolutionary algorithm typically failed to improve initial candidates with larger Levenshtein distances (11% of all spectra).

Overall, Spectralis-EA substantially increased the recall over the entire precision range against Casanovo and Novor (Fig. 5c). Moreover,

only 3% of initially correct peptides were corrupted after the evolutionary algorithm. Spectralis-EA reached a slightly lower recall at 90% precision than the rescoring of Casanovo-Novor candidates with Spectralis-score (40% vs. 43% median recall across samples, Fig. 5d). However, it significantly increased the overall recall consistently across samples (58% vs. 55% median recall across samples, Fig. 5e, Supplementary Fig. 19). Hence, the evolutionary algorithm led to minor improvements compared to rescoring Casanovo-Novor candidates. When investigating the Spectralis-EA limitations, we noticed that very few initial peptides with actual Levenshtein distance 2 could be improved (Fig. 5b). For these incorrect peptides, the Spectralis-score is very close to the Spectralis-score of their corresponding correct peptides (Supplementary Fig. 20). We reasoned that these peptides are difficult to discern by their experimental spectrum. In particular, distinguishing two peptides differing by a single permutation of two adjacent residues (thus at a Levenshtein distance of 2 from each other) is hard when the discriminative peak is missing. One advantage of the evolutionary algorithm is the ability to report an entire population of candidate peptides. Considering the two most highly scored predicted

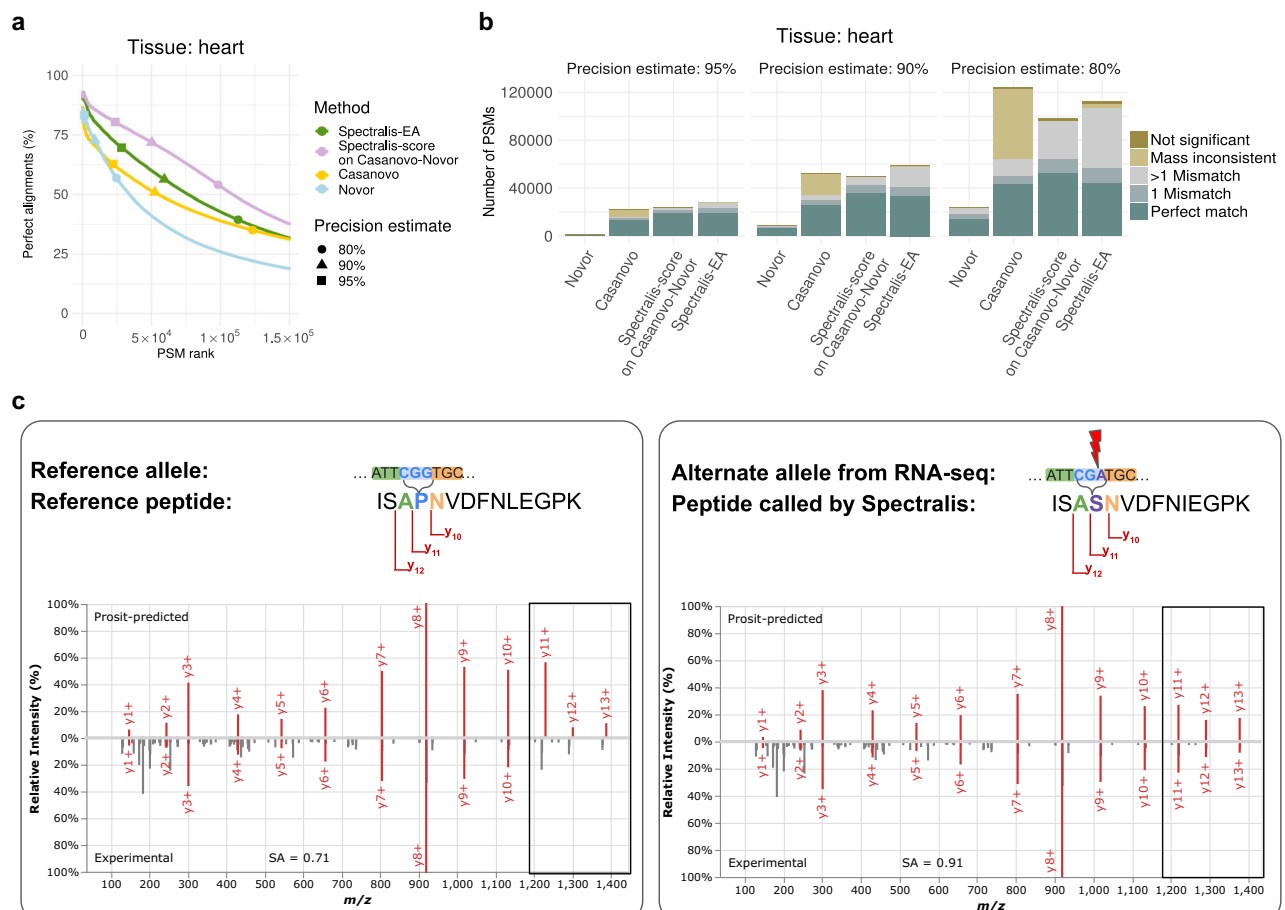

**Fig. 6 | Application to unidentified spectra and variant calling. a** Percentage of perfect alignments with mass consistent with the precursor *m/z* queried on peptide sequences by Novor, Casanovo, Spectralis-score on Casanovo-Novor and Spectralis-EA against a set of known and predicted gene translations using blastp on spectra not identified by MaxQuant of the heart sample, showing only the top 150,000 ranked candidate peptides of each method (out of 1,167,029). For clarity, the first 100 peptides are omitted. **b** Number of PSMs with perfect alignments, alignments with one mismatch, alignments with multiple mismatches, with mass not consistent with the precursor *m/z*, and without significant alignments for Novor, Casanovo, Spectralis-score on Casanovo-Novor and Spectralis-EA at

different precision estimates on the set of spectra of the heart sample unidentified by MaxQuant. **c** Left: Prosit-predicted spectrum (top) and experimental spectrum (bottom) for the reference peptide sequence ISAPNVDFNLEGPK. The cartoon illustrates the relevant nucleotide sequence and fragment ion series assuming the reference genome allele. Right: Same as for left, but for the peptide sequence ISASNVDFNIEGPK predicted by Spectralis-EA for the same experimental spectrum. Cartoon as in left inset for the alternative allele detected on RNA-seq of the same sample. A proline instead of serine is present at the fourth position. Parts of the spectra differing in left and right are shown in boxes. Source data are provided as a Source Data file.

peptides per spectrum, Spectralis-EA further increased the overall recall from 56 to 58% (median across samples) while lowering the recall at 90% precision from 40% down to 36% (median across samples, Supplementary Figs. 21–22). Altogether, these results indicate that Spectralis-EA allows the generation of several plausible candidates, a feature of particular interest when dealing with ambiguous spectra. However, for high-precision use cases, applying Spectralis-score on Casanovo-Novor candidates should be favored.

### Robustness of Spectralis on non-identified spectra and its applicability to variant calling

All Spectralis components were trained on a ground truth dataset of peptides identified by the database search algorithm MaxQuant. However, spectra confidently identified by MaxQuant tend to be of higher quality than other spectra, for instance, because they have fewer missing peaks[36,37]. Hence, it is important to assess whether Spectralis remains reliable on other spectra. To this end, we applied Spectralis to 1,167,029 spectra that were unidentified by MaxQuant and compared its performance to that on the 36,312 MaxQuant-identified spectra from the test set of the heart sample.

Assessing the performance on unidentified spectra is difficult due to the lack of ground truth. As a proxy, we considered the percentage of predicted peptides that perfectly align on the human proteome while giving no penalty to isoleucine-leucine mismatches (Methods). As expected, peptides identified by MaxQuant almost exclusively (99.9%) resulted in perfect alignments (Supplementary Fig. 23). On spectra not identified by MaxQuant, Spectralis consistently achieved a higher proportion of perfect alignments with a mass consistent with the precursor *m/z* compared to Casanovo and Novor, especially in the top-ranking predictions (Fig. 6a, Supplementary Fig. 23), where Spectralis-score applied to Casanovo-Novor candidates outperformed Spectralis-EA. Moreover, the percentage of perfect alignments remained high with 80% of perfect alignments at an estimated precision of 95%, and 72% at an estimated precision of 90% for Spectralis-score on Casanovo-Novor candidates (Methods). Similar patterns and performances were found for both spectra identified and not identified by MaxQuant in the brain sample (Supplementary Fig. 24).

A large percentage of Casanovo-Novor peptides ranked by Spectralis-score at a 90% precision estimate had a mass consistent with

the precursor *m/z* and aligned perfectly when allowing for a single mismatch (84%, Fig. 6b). Some of these mismatches are the result of missense mutations. One example is the correct call of a single amino acid variant in the AHNAK nucleoprotein in the heart sample (Fig. 6c). Spectralis predicted the peptide ISA**S**NVDFN[IL]EGPK which is ranked among the top 0.8% of the sequences, had a predicted Levenshtein distance of 0.18, and aligned to the reference peptide ISA**P**NVDFN-LEGPK. A proline instead of serine is present in the reference peptide at the fourth position. This missense variant was supported by RNA-seq data of the same sample (NM_001620.3:c.16348 C > T, p.Pro5450Ser, Methods). Using Spectralis, we were able to identify evidence of this genetic variant expressed in the heart sample directly from spectra independent of genetic data. Altogether, these results indicate that Spectralis can identify single amino acid changes in peptides and can be used to identify genetic variants indirectly from spectra.

## Discussion

In conclusion, we have introduced Spectralis, a method for de novo peptide sequencing that substantially increases sensitivity in the high precision range over state-of-the-art. Spectralis builds upon a modeling task, bin reclassification, which assigns ion series to discretized *m/z* values even in the absence of a peak. We showed that predicted bin classes enable improved scoring of PSMs. Using Levenshtein distance estimates as PSM scores, we demonstrated that rescoring peptides predicted by existing de novo peptide sequencing methods could improve recall by nearly two-fold at 90% precision. Furthermore, we devised an evolutionary algorithm leveraging these modeling innovations, resulting in an increased overall recall.

Our score does not improve the overall recall as it does not modify predicted peptides. However, it is very advantageous in practice to achieve a better separation between correctly and incorrectly predicted peptides. The scoring function can be used as a standalone method with little computational cost to rank. It also allows the comparison and integration of predicted peptides from several methods for de novo peptide sequencing using a single procedure. The guided mutations showed promising results for improving incorrect peptide candidates. However, the evolutionary algorithm leveraging guided mutations yielded modest improvements over rescoring alone. Nonetheless, we found that it allows generating several plausible candidates with very small Levenshtein distances to the correct peptide. Considering two or more high-confidence predictions per spectra could be useful for applications in which identifying a large portion of the peptide, yet not entirely, is of interest, for instance in cases of ambiguous spectra. For high-precision use cases, however, applying our score on candidate peptides proposed by existing de novo sequencing tools should be favored.

The graph used to derive guided peptide mutations is reminiscent of the widely used spectrum graph defined on peaks of an experimental spectrum[3]. A limitation of our graph representation compared to peak-based spectrum graphs is that we operate at the 1-Dalton resolution. Even though 1 Dalton corresponds approximately to the mass of a proton or a neutron, mass spectrometers allow measurements in higher resolutions which could in principle be leveraged. However, the 1-Dalton resolution is not a conceptual limit of our approach. Higher resolutions could be obtained at the cost of longer run time. The advantage of a bin-based rather than a peak-based graph, however, is that the nodes of our graph do not depend on the presence of an experimental peak, but only on the output of the bin reclassification. Therefore, this facilitates the generation of paths connecting nodes spaced by single amino acid masses. It should also be noted that Spectralis-score, which integrates Prosit predictions at a tolerance of 20 parts-per-million, leverages more highly resolved *m/z* ratio information.

We showed evidence of a rare missense variant with a maximum allele frequency smaller than 1%[38]. Thus, the ability to identify rare variants independent of genomic data lends credence to the idea that spectra contain personally identifiable information[39,40]. As de novo peptide sequencing continues to improve, we are getting closer to being able to re-identify individuals by means of mass spectrometry. Consequently, we agree that raw mass spectrometry proteomics data must be shared through data access portals with similar data control measures as next-generation sequencing data[41].

A newer version of Casanovo, Casanovo v3.2.0[42], has been developed concurrently with Spectralis. Casanovo v3.2.0 is a much-improved version of Casanovo v2.0.0 obtained by training on a very large dataset consisting of ~30 million PSMs. A revised version of Spectralis-score trained on Casanovo v3.2.0 scores still modestly, yet significantly increases recall at 90% precision on six of nine species (Supplementary Fig. 25). Future work, outside the scope of this study, is necessary to investigate further the complementary of the two approaches, e.g., by training the bin reclassification algorithms on the remaining errors of Casanovo v3.2.0.

One limitation of our study is that Spectralis is so far restricted to a single post-translational modification, methionine oxidation. Further post-translational modifications could be addressed in future work by extending the AA-gapped convolutions. For instance, modeling phosphorylation in animals would require adding the phosphorylated masses of three amino acids only.

Another limitation is that our approach assumes a single correct peptide per spectrum. To this end, we have restricted our database search ground truth to at most one peptide for each spectrum. However, studies have estimated that approximately half of all spectra are chimeric, i.e., they contain peaks from two or more precursor ions with similar masses and retention times[43–45]. This might further explain the limited overall recall of Spectralis and other earlier de novo peptide sequencing tools, which all also assume a single peptide per spectrum. Modeling mixtures of peptides would require different modeling schemes and the establishment of suitable ground truth data.

Despite these limitations, Spectralis exhibits strong de novo peptide sequencing performance especially at high precision ranges, allowing it to be used for variant calling. It could therefore make proteomics more amenable to applications ranging from pathogen surveillance to immuno-peptidomics and metaproteomics.

## Methods

### Datasets

We downloaded the publicly available dataset from Wang et al.[29] consisting of 7,902,759 experimental spectra across 30 healthy human samples from the PRoteomics IDEntifications (PRIDE) database with identifier PXD010154. We defined 302,054 peptide identifications by the database search engine MaxQuant (version 1.5.3.30)[28] at 1% FDR as a ground truth dataset of correct peptide identifications. For this, the spectra were previously searched against the Ensembl human proteome database (release-83, GRCh38)[46]. Carbamidomethyl (C) was specified as a fixed modification and methionine (M) oxidation and acetylation (Protein N-Term) were considered variable modifications. Trypsin/P was specified as the proteolytic enzyme with 2 maximum missed cleavages. In addition, we downloaded RNA-seq data corresponding to the same dataset from Wang et al.[29] from the ArrayExpress database, study E-MTAB-2836.

We downloaded the nine-species dataset first introduced by Tran et al.[25] consisting of about 1.5 million spectra across nine species. The ground truth dataset of peptide identifications was obtained from the FTP server of the MassIVE database (ftp://DeepNovo2017@massive.ucsd.edu) provided by Tran et al.[25]. As the data provided does not contain raw spectra but only preprocessed spectra, we obtained those from the original repositories of each species in PRIDE with identifiers: PXD005025, PXD004948, PXD004325, PXD004565, PXD004536, PXD004947, PXD003868, PXD004467, and PXD004424.

## Data preprocessing

To define the final set of ground truth peptide identifications and their corresponding experimental spectra for the human dataset by Wang et al.[29], we removed all correct peptides with a length larger than 30 and smaller than 7, as well as those with a charge larger than 6. Moreover, we removed peptides marked as decoys. We discarded secondary peptides ("MULTI-SECPEP") so that each experimental spectrum is attributed to at most one peptide sequence, namely the one with the highest score. Furthermore, we kept only unmodified peptides or peptides with methionine oxidation in the dataset. Finally, we removed all peptide sequences with a computed mass differing from the experimental mass by more than one Dalton. The experimental masses were derived from the original precursor $m/z$ and precursor charge which are contained in the raw files. Raw spectra were converted into MGF format using pyteomics[47] (https://github.com/levitsky/pyteomics, v4.6).

We ran Novor (v1.05)[16], Casanovo (v2.0.0)[27], DeepNovo (v0.0.1)[25], and PointNovo (v0.0.1)[26] on all spectra from the human dataset by Wang et al.[29]. We obtained one candidate peptide with its respective score for each spectrum and each tool. For Novor, we specified 10 ppm for the precursor mass tolerance and 0.05 Da for the fragment mass tolerance. Post-translational modification settings consisted of carbamidomethyl as a fixed modification and methionine oxidation as a variable modification. The fragmentation technique was set to HCD (Higher Energy Collision Dissociation) and the mass analyzer to FT (Fourier Transform). We ran Casanovo v2.0.0 and DeepNovo based on the publicly available pre-trained weights for each model trained on 8 different species excluding human peptides.

We split the human dataset by Wang et al.[29] consisting of unique PSMs with peptide identifications by MaxQuant, Novor, and Casanovo (v2.0.0) and our own mutations of correct peptide sequences by peptide into train (80%), validation (10%), and test (10%) set. On the train set, we removed peptide identifications whose computed mass differed from the mass derived from the experimental precursor $m/z$ by more than 20 ppm. Moreover, we obtained further incorrect candidate peptides for the train test by performing isobaric amino acid substitutions and permutations of adjacent amino acids on correct peptide identifications. We trained and evaluated all models using the same split.

For the nine-species dataset, we obtained peptide predictions from the raw spectra by running Novor (v1.05) and Casanovo (v2.0.0). In addition, we ran Casanovo (v3.2.0) on all unprocessed raw spectra. We obtained peptide sequences proposed by DeepNovo from the MassIVE repository (MSV000081382). To obtain peptide sequences by PointNovo, we retrained the models as described by Qiao et al.[26] on the nine-species dataset based on the publicly available code repository. For the evaluation of our methods, we removed all PSMs from the nine-species dataset for which the correct peptide was contained in the training set of the human dataset by Wang et al.[29] Moreover, for each species, we removed PSMs for which the correct sequences were contained in any other species. We also removed all PSMs for which the experimental mass derived from the precursor $m/z$ and precursor charge differed from the mass of the correct peptide by more than one Dalton.

## Bin reclassification with AA-gapped convolutions

The neural network for bin reclassification receives a discretized (binned) spectrum representation and predicts membership for singly charged b-ions and y-ions for each bin. We used a bin resolution of 1 Da and considered $m/z$ ratios up to 2000 Da. The following input features were computed for every bin: (i) the sum of all experimental peak intensities within the bin boundaries, (ii) the sum of peak intensities within the bin boundaries that Prosit[22] predicted for the input candidate peptide, and (iii) the binary class labels for b-ions and y-ions of the input candidate peptide.

The model consists of several AA-gapped convolution layers with skip connections between layers. We defined an AA-gapped convolution as a convolution operation where dilations correspond to the molecular masses of the 20 canonical amino acids and methionine oxidation. We used zero-padding to maintain the length dimension constant across layers. The number of filters was the same for all inner layers. Batch normalization and ReLU activations were applied after each convolution layer. We trained the model with Adam[48] using a batch size of 512 optimizing focal loss[49], and applied reduction of the learning rate on plateau as well as early stopping. Hyper-parameter search using optuna[50] allowed optimizing the number of filters per inner layer, the number of layers, the dropout rate, and the learning rate. The best model consisted of 16 AA-gapped convolution layers with 20 filters in each layer, a dropout rate of 0.3, and was trained with a learning rate of $4 \times 10^{-5}$. It was trained using PyTorch (v1.8.1)[47] on four A40 GPUs for 30 epochs, which resulted in a total training time of ~1.5 days.

## Deep learning-based guided mutations

We implemented a graph-based algorithm that generates additional peptide sequences for a given input sequence using the predicted bin probabilities for b-ions and y-ions by the bin reclassification model. We constructed a graph by introducing a node for every $m/z$ bin with a predicted probability larger than 0.35.

In order to deal with prefix fragments (b-ions) and suffix fragments (y-ions) in a unified fashion, we transformed $m/z$ values of predicted b-ions to their complement to the precursor $m/z$. We defined the maximum over the two predicted probabilities for the b-ions and y-ions as node weights.

In addition, we added a source node with $m/z$ value equal to 19 (i.e., the mass of one water molecule and one proton) and a target node with $m/z$ bin equal the discretized experimental peptide mass derived from the precursor $m/z$ and a proton. Source and target nodes received a node weight of one. Moreover, nodes for the $m/z$ bins corresponding to y-ions of the input sequence were introduced to the graph with a node weight of 0.01.

We allowed an edge between two nodes if the difference between the $m/z$ bins of the nodes corresponded to the discretized molecular mass of any amino acid. We labeled the edges with all amino acids that fulfilled the constraint.

To create additional peptide sequences, we performed weighted random walks starting from the source node until the target node was reached. To ensure that all random walks starting from the source node eventually led to the target node, we removed all nodes and edges that were not contained in any path from source to target. Edge probabilities for transition were defined based on the node weights. For any edge $e = (v, w)$ with node weights $p_v$ and $p_w$ for the nodes $v$ and $w$, we computed its edge probability $p_e$ as follows:

$$p_e := \frac{p_v + p_w}{\sum_{w',(v,w') \in E}(p_v + p_{w'})} \tag{1}$$

The peptide sequence can be recovered by concatenating all edge labels in the reversed path, thus starting from the target node. If more than one amino acid was labeled in an edge, one of them is selected at random.

## Scoring procedure

Spectralis-score of a PSM was estimated as the Levenshtein distance[33] of an input peptide sequence to the correct peptide sequence. The Levenshtein distance was computed with equal weights for insertions, deletions, and substitutions using the Python package editdistance (https://github.com/roy-ht/editdistance, v0.5.3). A random forest regressor served to predict the Levenshtein distance of a peptide sequence to its correct sequence.

We defined 114 features as input for the model derived from the comparison between Prosit-predicted and experimental spectra. To this end, we first applied base peak normalization to each experimental spectrum denoted as ($\mathbf{M}^{exp}$, $\mathbf{I}^{exp}$) consisting of $m/z$ value and intensity pairs and to each Prosit-predicted spectrum ($\mathbf{M}^{theo}$, $\mathbf{I}^{theo}$) consisting of $k \leq m/z$ value and intensity pairs. Intensity values below 0.02 were set to zero. Next, we defined experimental peaks, if any, corresponding theoretical peaks. The corresponding intensity $\hat{I}_k^{exp}$ to a theoretical intensity $I_k^{theo}$ of a peak $k$ in the theoretical spectrum was defined as the intensity $I_j^{exp}$ of its closest peak $j$ within a mass tolerance $\delta = 20$ ppm as follows:

$$\hat{I}_k^{exp} := \begin{cases} I_j^{exp}, & \text{if } |M_k^{theo} - M_j^{exp}| < \delta \cdot M_k^{theo} \\ 0, & \text{otherwise}. \end{cases} \quad (2)$$

We labeled these corresponding experimental peaks as b and y fragment ions according to the Prosit-predicted annotation.

We provided the model with three complementary feature types: similarity features, counting features, and features derived from the bin reclassification model.

The similarity features capture the quantitative agreement between Prosit-predicted and experimental peak intensities. They consist of the normalized spectral angle as defined earlier[22], the Pearson correlation coefficient, cosine similarity, as well as the mean, standard deviation, quantiles, maximum, and minimum of the absolute differences.

The counting features capture the qualitative agreement between Prosit-predicted and experimental $m/z$ ratios of peaks. They consist of the number (absolute and relative to the number of predicted peaks) of corresponding peaks for all four combinations of zero and nonzero experimental and theoretical intensities.

The similarity features and the counting features were generated for all peaks jointly, as well as for the b fragment ions and for the y fragment ions separately.

The features derived from the bin reclassification model were the amount of predicted bin class changes at various bin probability thresholds (0.25, 0.3, 0.35, 0.4, 0.45, and 0.5).

A list containing all features and computed feature importances obtained from the mean of the computed absolute SHAP values[51] is provided in Supplementary Table 1.

The random forest predictor was trained to predict $\log_2 (d + 1)$, where $d$ is the Levenshtein distance of the peptide, minimizing the sum of squared errors using scikit-learn (v0.24.2)[52]. The final model, selected after hyper-parameter search with optuna[50], contained 86 individual trees, a maximum tree depth of 175, a maximum number of 36 features for each node split, and a minimum amount of 112 samples per leaf node.

For comparison, an XGBoost (v1.6.2)[34] model was fitted with the same target variable and same features as the random forest using scikit-learn. The final XGBoost model, selected after hyper-parameter search with optuna, consisted of 410 gradient-boosted trees, a maximum depth of 10, a ratio of 0.17 for subsampling features when constructing each tree, and a ratio of 0.92 for subsampling the training dataset. All other hyper-parameters were set to the default values.

The score integrating Spectralis-score and the spectral angle between the spectrum predicted by PredFull[35] and the experimental spectrum was derived by fitting a logistic regression on these two scalars as features without any interaction term on the training dataset of the heart sample.

An alternative score was trained taking the defined 114 features as input, as well as the original scores provided by Casanovo v3.2.0 employing the leave-one-species-out cross-validation proposed by Tran et al.[25]. This score was obtained by fitting an XGBoost model to predict the Levenshtein distance of a candidate peptide to the correct peptide.

## Evolutionary algorithm

For each experimental spectrum, candidate peptides provided by any de novo peptide sequencing tool served as input for the evolutionary algorithm. Here, for each experimental spectrum, we considered the candidate peptide provided by Casanovo as the initial sequence only if the difference between the computed peptide mass and the experimental mass derived from the precursor $m/z$ was not larger than 1 Da. Otherwise, we started the optimization procedure with the candidate peptide generated by Novor. We denote this combination of sequences by Casanovo and Novor as Casanovo-Novor.

An initial population of candidate peptides was constructed by random isobaric substitutions and permutations of certain residues of the initial sequence: at most 3 consecutive residues were replaced by a combination of amino acids so that the total mass difference to the initial peptide sequence was not larger than 20 ppm.

At each generation, a set of $n$ peptides is selected for the next generation based on the Spectralis-score $s_1,...,s_n$ of candidate peptides in a current generation. The $n_e$ highest-scored peptides were directly inherited to the next generation. To maintain a fixed number of individuals in each generation, $j := n - n_e$ candidate peptides were selected for mutation in the next generation. For this, we assigned a weight for selection $w_i$ to each peptide with index $i$ and score $s_i$ as follows:

$$w_i := \exp\left(\frac{1}{T}(s_i - s^*)\right) \quad (3)$$

where $T$ denotes the temperature constant of the optimization procedure and $s^*$ the score of the fittest element in the current generation. After defining all selection weights, the selection procedure chose $j$ peptides for mutation according to these weights. On each of those $j$ peptides, we applied the guided mutation procedure.

Both selection and mutation procedures were repeated for $m$ generations, each of them with the same size of $n$ individuals and elite size of $n_e$, before the most highly scored candidate peptide according to the selected fitness function was returned as the final peptide sequence of the given spectrum. Hyper-parameter grid search on a random subset of the validation set identified $m = 5$, $n = 1,024$ and $n_e = 103$ as optimal hyper-parameters.

For initial peptide sequences of length larger than 30 or precursor charge larger than 6, we returned the initial sequence and the lowest possible score. Initial peptide sequences with an estimated Levenshtein distance smaller than 1 and larger than 7 were not optimized but returned unmodified as the final sequence of the evolutionary algorithm.

## Peptide alignment and variant calling

Peptide alignments were obtained by running blastp (version 2.12.0+)[53] against all translations from Ensembl genes and ab initio gene predictions provided by Ensembl human proteome database[46,54] genome build GRCh38, release 83. As a scoring matrix, we used the identity matrix, modified such that leucine and isoleucine were considered equivalent. All other blastp settings were set to their default values, including the value of 10 for the e-value. We restricted the output of blastp to at most one hit per queried peptide sequence. If multiple hits were returned by the search, we selected the hit with the lowest e-value. We defined a query peptide to be a perfect alignment if the peptide is identical to the target peptide except for differences between leucine and isoleucine.

For each method, we computed the score cutoff for three precision values (80, 90, and 95%) as the median across the 30 samples of the score cutoffs yielding these precision values on spectra identified by MaxQuant.

To call missense variants from the selected RNA-seq sample (RNA-seq ID: SAMEA2154361, corresponding proteomics ID: heart_5a), we first aligned the RNA-seq reads using STAR (v2.7.10a)[55] as part of the

nf-core rnaseq[56] module to the hg38 genome assembly using default parameters. We used GATK haplotypecaller through the RNA-seq variant calling module of the Detection of RNA Outliers Pipeline (DROP v1.2.2)[57] to call variants. Variants that were missense were identified using VEP v.106[54].

To map peptides to RNA-seq based missense variants, we ran a BLAST[53,58] search using tblastn which provided nucleotide coordinates for each peptide. The tblastn results were processed in a similar matter described for the blastp results above. We overlapped the nucleotide coordinates with the obtained missense variants from VEP using GenomicRanges[59].

### Evaluation metrics

On top of precision-recall at bin level, we further evaluated the model for bin reclassification with change precision and change recall curves, which use change probabilities instead of the original probabilities predicted by the model. For every bin, the change probability was defined as the predicted probability $p$, for bins with an initial label equal to 1, and $1 - p$ for bins with an initial label equal to 0.

We evaluated the de novo peptide sequencing methods with precision-recall curves at peptide level computed on the set of spectra identified with MaxQuant at 1% FDR. Peptide-level recall was defined as the fraction of correct peptide sequences over the total number of peptide sequences identified with MaxQuant at 1% FDR. Note that unlike recall defined for binary classifiers, peptide-level recall is not guaranteed to reach one at the most lenient score cutoff.

### Reporting summary

Further information on research design is available in the Nature Portfolio Reporting Summary linked to this article.

## Data availability

The mass spectrometric raw data from the human dataset by Wang et al. including the MaxQuant Spectronaut search data is available via the PRIDE database with the dataset identifier PXD010154. RNA-Seq data is available in the following database: ArrayExpress E-MTAB-2836. The human proteome database (genome build GRCh38, release 83) was downloaded from Ensembl (https://ftp.ensembl.org/pub/release-83/fasta/homo_sapiens/pep). The raw mass spectrometric data for the nine-species dataset by Tran et al. is available via the PRIDE database with identifiers: PXD005025, PXD004948, PXD004325, PXD004565, PXD004536,PXD004947, PXD003868, PXD004467, and PXD004424. The correct peptide identifications, as well as predictions by Deep-Novo, can be downloaded from the MassIVE repository with identifier MSV000081382. Model weights for running Casanovo were downloaded from Zenodo with DOI zenodo.6791263[60]. The trained bin reclassification model and random forest, as well as Novor, Casanovo, DeepNovo, PointNovo and Spectralis predicted peptides with respective scores are deposited on Zenodo with DOI zenodo.8393846[61]. The data to reproduce the main figures in this study have been deposited in the Figshare repository with DOI figshare.23536794[62]. Source data are provided with this paper as a Source Data file. Source data are provided with this paper.

## Code availability

Source code and scripts are available on GitHub at https://github.com/gagneurlab/spectralis[63].

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

## Acknowledgements

We thank Oğuz Gültepe for his contributions on the initial work of this project, as well as Vicente Yépez for comments on the manuscript. We thank Felix Brechtmann for suggesting modeling the Levenshtein distance and for many fruitful discussions, and Stefan Dvoretskii for using an evolutionary algorithm. We thank Alexander Karollus for the helpful discussions and for coming up with the name of the method. We thank Florian Hölzlwimmer for his considerate and talented support with the GPU infrastructure. Furthermore, we thank Wassim Gabriel and Ludwig Lautenbacher for their assistance with the client to obtain Prosit predictions and scoring features. The IBM infrastructure hosting Prosit is operated and maintained by the UCC at the TUM. This work is supported by the Bundesministerium für Bildung und Forschung (BMBF) through the project CLINSPECT-M (FKZ031L0214A to D.K., J.H., and J.G.), the Deutsche Forschungsgemeinschaft (DFG, German Research Foundation) via the project NFDI 1/1 "GHGA - German Human

Genome-Phenome Archive" (#441914366 to N.H.S.), and the European Union through the Horizon 2020 Program under Grant Agreement 823839 (H2020-INFRAIA-2018-1; EPIC-XS to M.W.). D.K., M.W. and J.G. were supported by a TUM Munich Data Science Institute (MDSI) seed fund.

## Author contributions

M.W. and J.G. jointly supervised the research. M.W. and J.G. conceived the method with the help of D.K. and J.H. D.K. and J.H. implemented the methods and performed the analysis on spectra identified by MaxQuant, with the help of Y.B. N.H.S. performed the peptide alignment and variant calling analysis with the help of Y.B. J.T. contributed to the method development. D.K., J.H., M.W., and J.G. wrote the manuscript with the help of N.H.S. and Y.B. All authors read and approved the manuscript.

## Funding

## Competing interests

M.W. is founder and shareholder of OmicScouts GmbH and MSAID GmbH, with no operational role in either company. The remaining authors declare no competing interests.
