## [Peer Review File · Nature Communications]

Reviewers' Comments:

Reviewer #1:

Remarks to the Author:

The authors present Spectralis, a novel framework for de novo peptide sequencing. Spectralis includes two key components: a bin correction model and a distance estimation model which can be used as an alternative to confidence estimation and evolutionary search.

The paper is well written with all the essential details. The methods and ideas are quite intriguing. This method has the potential to be able to help improve the prediction of other de novo models.

I have some questions/comments.

(1) AA-gapped convolution

It's a neat idea to use dilate convolution in order to constrain the model to look at the important locations based on the mass. This gap idea is presented in other de novo works such as stacking inputs shifted by the amount of mass. Dilate convolution is quite notorious for its slowness and inefficiency. Have you considered comparing with other techniques in order to accomplish the same goal?

The model uses dilate convolution throughout the entire model. But, I believe only a few dilate layers should suffice in order to focus the model to the correct locations. Have you consider replacing the upper layers with regular CNNs?

(2) Distance prediction

I am curious about the generalizability of the trained distance prediction model. Would it perform well on sequences predicted by other de novo models?

In the same manner, I wonder if the model behave similarly on different generations of the Spectralis-GA. As the generations mutate they should be further and further from the training data, which might cause worse estimation performance.

(3) minor

- Is there a reason why Fig. 4(B) is in the log-log scale rather than a normal 0-10 bin.

- Any comments on the choice of random forest? Rather than let's say XGBoost.

Reviewer #2:

Remarks to the Author:

The paper presented a deep learning approach to classify m/z bins to correspond to singly charge b/y ion masses, and exploited the prediction to improve de novo peptide sequencing algorithms such as Casanovo. The proposed method seems to be similar to the early approaches to de novo sequencing that scores putative precursor masses based on fragmentation patterns (i.e., intensities of b/y ions and neutral loss ions), such as the algorithms adopted by PepNovo (based on Bayesian network) and NovoHMM (based on Hidden Markov Models), and the methods for spectral alignment and spectral network. The method proposed here is based on deep neural networks and thus may capture complex correlations among fragment ions. However, I think the authors should connect their approach with the early works in computational proteomics as laid out above. Moreover, the paper present the the improvement of the de novo sequenced peptides on the sequencing results from Casanovo. A straightforward approach is to directly output the peptide from the bin classification results, e.g. using a dynamic programming algorithm (i.e., to find the longest path in the spectrum graph in which the vertices are weighted by the bin classification score, like the algorithm adopted by PepNovo). If the bin classification results are sufficiently accurate, this baseline approach should offer good sequencing results. The post-processing approach presented in the current manuscript seems to limited to Casanovo, and it is unclear if the improvement can be achieved on the other de novo sequencing algorithms.

I have also a few other suggestions regarding the evaluation of the methods.

1. The authors should include PointNovo into the benchmarking. PointNovo shows significant improvement over other published de novo sequencing algorithms.
2. The author should clarify if there is overlapping peptide/spectra in the training data for the bin classification model, and the testing data for the de novo sequencing and for PSM scoring, to avoid information leakage.
3. The PSM scoring based on bin classification should be benchmarked against the scoring between the experimental and predicted spectra (from the peptide of the PSM) by the spectral prediction models such as Prosit (for b/y ions) and PredFull (for whole spectra).

Reviewer #3:

Remarks to the Author:

The proposed Spectralis is a new de novo peptide sequencing method for peptide tandem mass (MS/MS) spectra based on the correction of de novo sequences output by other methods, with the specific approach described in the manuscript proposing to improve on the sequencing accuracy of Casanovo. The final results were also compared with Novor. The proposed approach is novel in the field and could provide a foundation for future improvements using related models. This method consists of three major components which are Bin reclassification, Spectralis-score and Spectralis-GA.

The bin reclassification proposes a modified convolutional layer called "AA-gapped convolution layer" introducing the idea of having dilations equal to the mass of amino acids and outputting probabilities that a b-ion or a y-ion are located at each bin. One minor downside of the current implementation is that it treats the spectrum masses (or m/z, to be more precise) as integer values, which loses some resolution and mass accuracy, but this could be likely be addressed by using more and smaller spectrum bins.

The proposed Spectralis-Score is a random forest regressor model to estimate the Levenshtein distance between the correct peptide and a peptide generated by any de novo sequencing algorithm based on 114 different features, some of which also derived from the bin reclassification results. Application of this approach led to significant improvements to the Recall results of both Casanovo and Novor de novo results at 90% precision. The results do show that the Levenshtein regression somewhat underestimates the real distance (especially for Novor), but this did not seem to significantly impact the reported refinement results based on Casanovo results.

The proposed direction of using a genetic algorithm for de novo sequencing (Spectralis-GA) uses the Spectralis-Score as the scoring function along with a guided mutation strategy with Casanovo-Novor result as the initial sequence. While the reported results show a minor improvement in overall Recall, this was not the case for the most important subset of the results at 90% precision. It is also important to note that the proposed genetic algorithm does not define a crossover operation so it is more like a random walk or a simulated annealing approach than a genetic algorithm approach, and the manuscript and the name of the approach should be adjusted accordingly.

While the authors did use appropriate amounts of data in the proper ways to train and evaluate the models, there were some choices of what to present that make it difficult to properly evaluate and compare the results with other approaches. First, it was not clear whether the results derived from the analysis of heart/brain data were representative of the results on all tissues. It could have been helpful to also include at least precision/recall curves for the other tissues (or at least also for the worst-performing tissue) in supplementary materials. Second, it would also have been useful to show precision/recall results partitioned by peptide length and charge state, since it is likely that the performance of the approach will vary based on these factors. Third, the manuscript should also include results on the same nine species dataset that was introduced by Tran et al. (ref 24 in current submission) and that was also used in the Casanovo preprint (doi:

<https://doi.org/10.1101/2022.02.07.479481>) to make it possible to assess how the results compare across the various approaches when the same data is analyzed by the authors of each approach.

Minor comments:

- The proposed PSM Scoring function should also be compared with other state-of-the-art scoring functions used for database search, since those may yield better sensitivity at the same false discovery rate.
- The reported results are based on an older version of Casanovo, not on the current version 3 – it would be good to confirm that the reported improvements still hold in relation to the current version.

Response to reviewers for “NCOMMS-23-02312-T”

We thank the reviewers for their helpful comments, which have helped us to strengthen the key points and improve the manuscript substantially. The most important changes and additional analyses we have conducted are:

- As requested by the reviewers, we have applied our confidence score for peptide-spectrum matches to peptide identifications by another de novo sequencing tool on the same dataset as presented in the manuscript. Moreover, we have applied the score to the nine-species dataset introduced by Tran et al., which is widely used in the field of de novo sequencing. These new results confirm the added value of Spectralis.
- We benchmarked our confidence score against the spectral angles given from the comparison between experimental and predicted spectra by the deep learning tools Prosit and PredFull. In these new evaluations, Spectralis-score remains the best-performing score.
- We compared our model for fragment ion series labeling based on a novel amino acid-gapped convolutional layer to models based on convolutional layers with and without regular dilations and a hybrid approach consisting of both amino acid-gapped and regular convolutions.
- We extended the evaluation of our methods by providing the performance of the best-performing and worst-performing tissues in the selected dataset and by stratifying the analysis by peptide length and precursor charge state.

In the following, we present our response to the reviewers' comments. The comments are in underlined italics, and the responses are in **blue regular font**. In each response, we refer to the subsections in the manuscript that have been adapted accordingly.

Reviewer #1 (Remarks to the Author):

The authors present Spectralis, a novel framework for de novo peptide sequencing. Spectralis includes two key components: a bin correction model and a distance estimation model which can be used as an alternative to confidence estimation and evolutionary search.

The paper is well written with all the essential details. The methods and ideas are quite intriguing. This method has the potential to be able to help improve the prediction of other de novo models.

I have some questions/comments.

(1) AA-gapped convolution

It's a neat idea to use dilate convolution in order to constrain the model to look at the important locations based on the mass. This gap idea is presented in other de novo works such as stacking inputs shifted by the amount of mass. Dilate convolution is quite notorious for its slowness and inefficiency. Have you considered comparing with other techniques in order to accomplish the same goal?

Response:

The reviewer is correct that convolutional neural networks (CNNs) have been proposed in the field of de novo sequencing with and without constraining the models to look at positions spaced by amino acid masses. In contrast to the idea of stacking inputs shifted by the amount of mass, as in Karunratanakul et al. (2019), our AA-gapped convolutions apply this constraint throughout all layers and not only for the input layer. We now explicitly make this distinction in the subsection "Overview of Spectralis".

Even though their implementations have improved over the last years, dilated convolutions remain slower than regular CNNs. We have now tried to achieve the same performance in terms of precision-recall at bin level by training CNNs with and without dilation with small (e.g. 5 and 7) and also with big filter sizes (e.g. corresponding to the discretized mass of the heaviest amino acid) to capture larger mass-to-charge differences between two peaks. As a representative sample, we used the heart sample (see the response to reviewer #3 and the subsection "Overview of Spectralis") to conduct these experiments. These new experiments showed that the performance of the model notably increased when using AA-gapped convolutions. We appended the evaluation of these different approaches to the Supplementary Material (Supplementary Fig. 3) and added a sentence to the subsection "Bin reclassification allows generating improved candidate peptides" of the manuscript.

The model uses dilate convolution throughout the entire model. But, I believe only a few dilate layers should suffice in order to focus the model to the correct locations. Have you consider replacing the upper layers with regular CNNs?

Response:

We thank the reviewer for their suggestion. We trained a version of the bin reclassification model which consists of AA-gapped convolutions for the first layers and regular convolutional layers for the upper layers. However, the final model that we presented in the manuscript

consisting of only AA-gapped convolutions substantially outperformed the hybrid approach. We show the comparison of both approaches in Supplementary Figure 3 and adapted the subsection "Bin reclassification allows generating improved candidate peptides" of the manuscript accordingly.

(2) Distance prediction

I am curious about the generalizability of the trained distance prediction model. Would it perform well on sequences predicted by other de novo models?

Response:

In addition to sequences predicted by Novor and Casanovo, we have now applied Spectralis-score to sequences proposed by DeepNovo on the dataset from Wang et al. (2019). The evaluation shows that, when compared to the original score, our score achieves higher recall at all precision ranges. The evaluation was added to the Supplementary Material (Supplementary Fig. 12) and the subsection "Levenshtein distance estimate improves PSM scoring" was adapted accordingly.

In the same manner, I wonder if the model behave similarly on different generations of the Spectralis-GA. As the generations mutate they should be further and further from the training data, which might cause worse estimation performance.

Response:

To address this reviewer's comment, we have evaluated the Levenshtein distance estimator for candidates generated during different generations of Spectralis-GA, which we have now renamed to Spectralis-EA as suggested by reviewer #3. We found that there is no decrease in the performance of the Levenshtein distance estimator for later generations of the algorithm. We added this evaluation to the Supplementary Material (Supplementary Fig. 15) and adapted the subsection "An evolutionary algorithm increases the sensitivity of de novo peptide sequencing" of the manuscript accordingly.

(3) minor

- Is there a reason why Fig. 4(B) is in the log-log scale rather than a normal 0-10 bin.

Response:

As the distribution of both actual and predicted Levenshtein distances is centered around small Levenshtein distances, we achieved a better visualization of all data points in logarithmic scale. For comparison, we attach the version of Fig. 4B in natural scale below.

- Any comments on the choice of random forest? Rather than let's say XGBoost.

Response:

Previously, we trained logistic regression models, simple fully connected neural networks, and support vector machines to assess the quality of a peptide-spectrum match. These models underperformed the random forest which estimates the Levenshtein distance of any peptide to the correct peptide. XGBoost is indeed often nowadays proposed as a go-to model for supervised learning. We have now trained and evaluated 20 XGBoost models and found similar performance as the random forest. We included this evaluation in the Supplementary Material (Supplementary Fig. 6) and added a sentence on this topic to the subsection "Levenshtein distance estimate improves PSM scoring" of the manuscript.

Reviewer #2 (Remarks to the Author):

The paper presented a deep learning approach to classify m/z bins to correspond to singly charge b/y ion masses, and exploited the prediction to improve de novo peptide sequencing algorithms such as Casanovo. The proposed method seems to be similar to the early approaches to de novo sequencing that scores putative precursor masses based on fragmentation patterns (i.e., intensities of b/y ions and neutral loss ions), such as the algorithms adopted by PepNovo (based on Bayesian network) and NovoHMM (based on Hidden Markov Models), and the methods for spectral alignment and spectral network. The method proposed here is based on deep neural networks and thus may capture complex correlations among fragment ions. However, I think the authors should connect their approach with the early works in computational proteomics as laid out above.

Response:

We now extended the introduction to better connect our approach to early works in the field of de novo sequencing. We were referring to PepNovo and now make the connection more explicit at the end of the introduction. Furthermore, we now refer to NovoHMM.

Moreover, the paper presents the improvement of the de novo sequenced peptides on the sequencing results from Casanovo. A straightforward approach is to directly output the peptide from the bin classification results, e.g. using a dynamic programming algorithm (i.e., to find the longest path in the spectrum graph in which the vertices are weighted by the bin classification score, like the algorithm adopted by PepNovo). If the bin classification results are sufficiently accurate, this baseline approach should offer good sequencing results.

Response:

The reviewer rightly points out that a straightforward approach to obtaining a new peptide candidate from the bin reclassification model is to use a dynamic programming algorithm on the constructed graph. Unfortunately, the bin classification results are not sufficiently accurate for this approach to work out of the box. This has been the reason for us to develop a score and Spectralis-EA. We now provide further analyses to make this point clear. Specifically, for each spectrum in the heart sample of the dataset introduced by Wang et al., we searched for the longest path in the weighted graph resulting from bin probabilities by the bin reclassification model. However, the correct sequence is often not the one corresponding to the path with the highest sum of bin probabilities. Therefore, the final peptide recall obtained from this dynamic programming approach is lower than the one obtained by Spectralis-GA, which we have now renamed to Spectralis-EA as suggested by reviewer #3. We added a sentence to the subsection "Bin reclassification allows generating improved candidate peptides" of the manuscript and appended the comparison of peptide recall by the different approaches to the Supplementary Material (Supplementary Fig. 4).

The post-processing approach presented in the current manuscript seems to limited to Casanovo, and it is unclear if the improvement can be achieved on the other de novo sequencing algorithms.

Response:

In our work, we propose two post-processing approaches that can be used separately: a rescoring procedure (Spectralis-score) and a fine-tuning procedure (Spectralis-EA). In the first submission of our manuscript, we evaluated Spectralis-score on sequences predicted by Novor and Casanovo separately and Spectralis-EA on the combination of sequences by Casanovo and Novor (Casanovo-Novor). We have now added evaluations of Spectralis-score for another de novo sequencing algorithm, DeepNovo (Supplementary Fig. 12), and another dataset (Tran et al. (2017), Supplementary Fig. 13). We did not consider doing this for Spectralis-EA since our largest improvement is on the score. The subsection "Levenshtein distance estimate improves PSM scoring" was extended in the manuscript accordingly.

I have also a few other suggestions regarding the evaluation of the methods.

1. The authors should include PointNovo into the benchmarking. PointNovo shows significant improvement over other published de novo sequencing algorithms.

Response:

We agree with the reviewer that PointNovo shows substantial improvement over other published de novo sequencing algorithms and slightly worse performance than Casanovo, as shown by Yilmaz et al. Furthermore, it is important to show the broad applicability of our method on other de novo sequencing algorithms. We tried running PointNovo on the selected dataset. However, the GitHub repository of PointNovo is not accessible and the model weights were not provided by the authors. We reached out to the authors but, unfortunately, we did not get a reply. Therefore, we applied Spectralis-score to sequences predicted by DeepNovo and showed that our score outperforms the original scores provided by this tool. Overall, we have shown the benefit of Spectralis-score on the three tools Novor, DeepNovo and also Casanovo, which outperforms PointNovo. We have adapted the subsection "Levenshtein distance estimate improves PSM scoring" of the manuscript and appended the evaluation to the Supplementary Material (Supplementary Fig. 12).

2. The author should clarify if there is overlapping peptide/spectra in the training data for the bin classification model, and the testing data for the de novo sequencing and for PSM scoring, to avoid information leakage.

Response:

We have now improved the wording in Methods. As potential information leakage is critical and as ensuring this does not happen is not trivial, we are now also describing our split strategy in the Results subsection "Overview of Spectralis". The selected split of train, validation and test data ensures that neither experimental spectra nor correct peptides are shared between train, validation and test set. We use the same split of train, validation and test set for all methods, i.e. bin reclassification model, PSM scoring and evolutionary algorithm. This means that the test set was never seen by the bin reclassification or random forest for PSM scoring during training and hyper-parameter tuning but was only used for the final evaluation.

3. The PSM scoring based on bin classification should be benchmarked against the scoring between the experimental and predicted spectra (from the peptide of the PSM) by the spectral prediction models such as Prosit (for b/y ions) and PredFull (for whole spectra).

Response:

We have now performed such analyses (Supplementary Fig. 8). To benchmark Spectralis-score against spectral prediction models such as Prosit and PredFull, we first aggregated the comparison between predicted and experimental spectra into one single scalar given by the spectral angle. Our PSM scoring achieves higher precision and recall than the spectral angles

by Prosit and PredFull across all ranges. Prosit-based spectral angle was already a feature of Spectralis-score. To assess whether PredFull provides complementary information, we have further fitted a logistic regression on Spectralis-score and PredFull-based spectral angle using the heart sample as a representative sample. This combined score did not lead to an increase in performance compared to only utilizing Spectralis-score. These results are shown in the new Supplementary Figure 8 and the subsection "Levenshtein distance estimate improves PSM scoring" was adapted accordingly.

Reviewer #3 (Remarks to the Author):

The proposed Spectralis is a new de novo peptide sequencing method for peptide tandem mass (MS/MS) spectra based on the correction of de novo sequences output by other methods, with the specific approach described in the manuscript proposing to improve on the sequencing accuracy of Casanovo. The final results were also compared with Novor. The proposed approach is novel in the field and could provide a foundation for future improvements using related models. This method consists of three major components which are Bin reclassification, Spectralis-score and Spectralis-GA.

The bin reclassification proposes a modified convolutional layer called "AA-gapped convolution layer" introducing the idea of having dilations equal to the mass of amino acids and outputting probabilities that a b-ion or a y-ion are located at each bin. One minor downside of the current implementation is that it treats the spectrum masses (or m/z, to be more precise) as integer values, which loses some resolution and mass accuracy, but this could be likely be addressed by using more and smaller spectrum bins.

Response:

We agree with the reviewer that increasing spectrum bin count and thus resolution could improve the performance of the bin reclassification model. However, this increases runtime, and we found performance to suffice for rescoring given the resolution of one Dalton per bin. Furthermore, by leveraging Prosit predictions, we integrate more fine-grained intensity information in the binned representation, alleviating the need for a high-resolution spectrum representation. As this reviewer correctly points out, the 1-Da resolution is not a conceptual limit of our approach and higher resolution could be obtained at the cost of longer run time. We extended and emphasized this point in the discussion of our manuscript.

The proposed Spectralis-Score is a random forest regressor model to estimate the Levenshtein distance between the correct peptide and a peptide generated by any de novo sequencing algorithm based on 114 different features, some of which also derived from the bin reclassification results. Application of this approach led to significant improvements to the Recall results of both Cananovo and Novor de novo results at 90% precision. The results do show that the Levenshtein regression somewhat underestimates the real distance (especially for Novor), but this did not seem to significantly impact the reported refinement results based on Casanovo results.

The proposed direction of using a genetic algorithm for de novo sequencing (Spectralis-GA) uses the Spectralis-Score as the scoring function along with a guided mutation strategy with Casanovo-Novor result as the initial sequence. While the reported results show a minor improvement in overall Recall, this was not the case for the most important subset of the results at 90% precision.

Response:

We thank the reviewer for their appreciation of our work.

It is also important to note that the proposed genetic algorithm does not define a crossover operation so it is more like a random walk or a simulated annealing approach than a genetic algorithm approach, and the manuscript and the name of the approach should be adjusted accordingly.

Response:

This reviewer correctly points out that genetic algorithms typically require a cross-over operation, which we did not develop. A more accurate categorization of our algorithm is as an Evolutionary Algorithm (EA, Katoch et al. (2021)). Evolutionary algorithms differ from simulated annealing, an optimization algorithm based on random walks, by operating on a population of candidates in parallel. In contrast, simulated annealing optimizes a single candidate at the time. We have now renamed Spectralis-GA as Spectralis-EA throughout the manuscript.

While the authors did use appropriate amounts of data in the proper ways to train and evaluate the models, there were some choices of what to present that make it difficult to properly evaluate and compare the results with other approaches.

First, it was not clear whether the results derived from the analysis of heart/brain data were representative of the results on all tissues. It could have been helpful to also include at least precision/recall curves for the other tissues (or at least also for the worst-performing tissue) in supplementary materials.

Response:

As the performance differs for all de novo sequencing tools across tissues, probably due to the different quality of the underlying samples, we felt that providing common precision-recall curves by pooling results across tissues could be misleading. We had therefore chosen to show precision-recall curves for the heart sample (in the main figures) and brain sample (in the Supplementary Material) as representative tissues and showed with boxplots the distribution of summary statistics such as recall at 90% precision or relative improvement. We feel that showing precision-recall curves for all 30 tissues would unnecessarily blow up the supplements. We could do it if this reviewer requests it. As an alternative, we have now appended precision-recall curves for the three best-performing and three worst-performing tissues to evaluate Spectralis-score and Spectralis-EA to the Supplementary Material (Supplementary Figs. 9, 15). These plots show that the heart and brain samples are not

cherry-picked examples and that our relative improvements are consistent across tissues. Moreover, we have added a sentence explaining this strategy of showing performance on representative tissues in the subsection "Overview of Spectralis".

Second, it would also have been useful to show precision/recall results partitioned by peptide length and charge state, since it is likely that the performance of the approach will vary based on these factors.

Response:

We have now computed precision-recall curves partitioned by both charge state and peptide length (new Supplementary Figs. 10,11). Our evaluation shows that we consistently improve the recall at all precision ranges compared to the original scores by both Novor and Casanovo, even though, as correctly suspected by this reviewer, there is a performance decrease in both the original score and Spectralis-score for higher charge states and larger peptides. The section "Levenshtein distance estimate improves PSM scoring" of the manuscript was adapted accordingly.

Third, the manuscript should also include results on the same nine species dataset that was introduced by Tran et al. (ref 24 in current submission) and that was also used in the Casanovo preprint (doi: <https://doi.org/10.1101/2022.02.07.479481>) to make it possible to assess how the results compare across the various approaches when the same data is analyzed by the authors of each approach.

Response:

We have now included results on the nine-species dataset (Supplementary Fig. 13). Here, we consider the same spectrum identifiers and "ground truth" peptides from PEAKS DB as Tran et al. (2017). Spectralis-score achieves a better performance on seven out of the nine species, on one species (PXD004948) the recall remains very similar at all precision ranges, and on the last species (PXD004536) the recall improves over the original score for precision ranges below 80%. These results are now reported in the subsection "Levenshtein distance estimate improves PSM scoring".

Minor comments:

- The proposed PSM Scoring function should also be compared with other state-of-the-art scoring functions used for database search, since those may yield better sensitivity at the same false discovery rate.

Response:

Scoring functions from database search that make use of reference genomes (e.g. via target-decoy strategies, Percolator) cannot be applied in the de novo peptide sequencing context. Moreover, current state-of-the-art in peptide identification suggests that incorporating predictions such as fragment intensities into the scoring functions yields significantly better sensitivity and specificity in comparison to other common database search engine scoring

functions such as Andromeda and XCorr that notoriously fail on differentiating isomers (Zolg et al. (2021), Declercq et al. (2022), Cormican et al. (2022)). Therefore, we have addressed this point by benchmarking Spectralis-score (which was inspired by the superior performance of a score combining multiple features) against the spectral angles between experimental and predicted spectra from both Prosit and PredFull as two possible applicable standalone scoring functions. We have appended this evaluation to the Supplementary Material (Supplementary Fig. 8) and adapted the subsection "Levenshtein distance estimate improves PSM scoring" accordingly. These results further confirm the added value of the Spectralis-score and are consistent with the fact that a combined score outperforms single scores.

- The reported results are based on an older version of Casanovo, not on the current version 3 – it would be good to confirm that the reported improvements still hold in relation to the current version.

Response:

Casanovo v3.2.0 is a much-improved version of Casanovo v2.0.0 obtained by training on a much larger dataset (around 30 million PSM versus 1.5 million before). Casanovo v3.2.0 has been developed concurrently with our work and co-submitted to this journal. Therefore, this study cannot be fairly reviewed against it. We have nevertheless trained and tested a version of Spectralis-score that takes the original features from Prosit and bin reclassification, as well as the original scores provided by Casanovo (v3.2.0) on the nine-species dataset with a leave-one-species-out cross-validation strategy. This score can modestly improve the recall at 90% precision over the initial recall by Casanovo (v3.2.0) on seven of nine species. Future work, outside the scope of this study, is necessary to further investigate the complementarity of the two approaches. We are now making that point in the Discussion and added this evaluation in a new supplementary figure (Supplementary Fig. 21).

Reviewers' Comments:

Reviewer #2:

Remarks to the Author:

The authors have improved the manuscript significantly improved their manuscript, and addressed most of my concerns. However, I have a few remaining concerns.

1. Comparison with PointNovo. The authors said PointNovo does not provide the model parameters, and thus it is not easy for a straightforward comparison. They further claimed Casanovo performs better than PointNovo. However, from the current version of Casanovo manuscript, it is hard to tell the performance gaining in most of the tested cases as it only used the publish results when prec-recall of PointNovo, while the casanovo results were shown with different prec/recall. Ideally, the performance comparison should be compared using the prec/recall curve. In fact, Pointnovo authors have provided the training code and training dataset, which make it easy to reproduce the model parameters. In addition, to make it a fair comparison, it would also be useful to investigate if different training data used by different models will impact the performance of de novo sequencing algorithms.

2. The paper shows that the rescoring PSMs using Spectralis-score improves the recall of de novo sequencing algorithms. I wonder if it is due to the re-rank of the candidate peptides sequenced on the same spectra, or the adjustment of the score distributions among the best PSMs of different spectra. If it is the former case, on what percentage of the cases, the improvement is due to swap of adjacent amino acid residues? If it is the latter case, how is the improvement correlated with the length of the sequenced peptides?

3. Can the model to predict the Spectralis-score be extended to predict the score each positions in the sequenced peptides? It seems the features contains sufficient information for such prediction, while the positional score will give the indication of the accuracy of each sequenced residue.

Reviewer #3:

Remarks to the Author:

The revised version of the manuscript was responsive to the reviewers' comments for the first submission, and the revised content significantly improves the presentation and characterization of the proposed approach. However, the improved characterization of Spectralis also revealed important new concerns that need to be addressed.

First, the new results for the 9 tissues datasets show a much lesser degree of improvement for Spectralis over Casanovo for a fair number of species, which brings into question the generalizability of the Spectralis models. Furthermore, it was not clear from the text whether the evaluation on the 9 tissues dataset removed all Peptide Spectrum Matches (PSMs) for all precursors (i.e., distinct pairs of modified sequence and charge state) which were included in the Spectralis training set due to sequence conservation between human and the other species. If these precursors were not excluded from the evaluation set then this should be revised to evaluate the performance of Spectralis without these precursors on the 9 species dataset. The revised version of the manuscript should include Spectralis predictions (as with the original submission for the PXD010154 dataset), as well as including columns for the spectrum files, scan numbers and ground truth identifications for all the PSMs used for the evaluation.

Second, the concerns with generalizability led the reviewer to a closer inspection of the data submitted to Zenodo, which revealed two new substantial concerns when cross-referenced with the MaxQuant identifications submitted with the original PXD010154 dataset whose data was used for both training and evaluating the models presented in this manuscript.

The first concern is that the total number of PSMs identified in human heart tissue by Maxquant for 4,033 precursors is higher than the number of PSMs for the same precursors reported in the Spectralis predictions, which leaves serious concerns that the remaining PSMs may have somehow been included in the training datasets, which would have been a serious problem with the

training/evaluation procedures (as the authors correctly acknowledged in the response to reviewer #2). As shown in the P013201_heart_precursors_reused.txt file provided with the review, there are many precursors with hundreds of PSMs in the same files in the original dataset for which there are also a subset of PSMs and predictions in the Spectralis set of reported results, which should not have been possible if the evaluation was performed as described in the manuscript and as replied to reviewer #2.

The second concern is that a direct comparison of the Spectralis predictions submitted to Zenodo with the MaxQuant identifications submitted with the original PXD010154 dataset yields a much worse Precision/Recall curve than what is reported in the manuscript (e.g., Figure 5c). As shown in the P013201_heart_spectralis_vs_maxquant.txt file provided with the review, the highest Recall observed at Precision 90% is Recall 41.3%, which appears to be lower than the values depicted in Figure 5c.

The most likely explanation for both of these concerns is that the authors likely used a different set of MaxQuant identifications than what was reported in the original dataset submission, possibly by refining those identifications with additional filters or criteria that were not clear in the manuscript. The simplest way to address these concerns would be to explain the differences by adding the following to the Zenodo data: (a) submit the full set of MaxQuant identifications used for training the models and (b) add columns to the Spectralis results submitted to Zenodo to also show the ground truth MaxQuant identifications that are being used to evaluate the performance of the models.

Finally, the evaluation of the data submitted with the manuscript also prompted an attempted evaluation of the code submitted with the manuscript, which actually revealed that the code does not work and is not usable even by developers, let alone by biologists or mass spectrometrists attempting to use the proposed approach. While the given link to the author's GitHub repository is accessible, there was no detailed description about the tool inside the repository or in the manuscript. More detailed inspection of the code (which is also almost completely undocumented) further revealed that the shared code relies on the ProSIT spectrum predictor which is accessed through remote procedure calls using certificate files. As such, even if the code worked (which it did not), it appears like users would not be able to use Spectralis as a standalone tool, and may not even be able to use it free of charge for their own data. While this is not necessarily a reason to prevent publication of the proposed approach, it does significantly reduce the potential impact of the proposed approach.

Response to reviewers for “NCOMMS-23-02312-A”

We hereby submit the revised version of our Nature Communications submission NCOMMS-23-02312A. We have addressed all new points raised by the reviewers. A point-by-point response can be found below. The most important changes and additional analyses are:

1. We have fully addressed reviewer #3's concerns which arose from comparing the data submitted to Zenodo to the original ProteomeXchange datasets by submitting the full set of MaxQuant identifications used for training and evaluating the models. To avoid confusion in the precision-recall calculations, we have expanded and improved the wording on the Method's section data processing.
2. We have addressed the issues that reviewer #3 identified regarding code usability by eliminating the dependency of our tool on the previous Prosit client, which was originally accessed through remote procedure calls using certificate files. Furthermore, we now provide installation and usage instructions, as well as an example data file for testing purposes in our public GitHub repository. The better code documentation, together with the additional information provided in GitHub, improves the usability of Spectralis.
3. As requested by reviewer #2, we have applied our confidence score to peptide identifications by PointNovo on the nine-species dataset introduced by Tran et al., and on the human dataset by Wang et al. to confirm the added value of Spectralis on a different de novo sequencing tool.

Altogether, we thank the reviewers for their constructive feedback and careful inspection of our manuscript, code repository and data submission. However, we would like to mention that, surprisingly, several points of this second revision round could have been raised in the first revision round. We hope that the next revision round focuses on open points so that we can swiftly converge to a decision.

In the following, we present our response to the reviewers' comments. The comments are in underlined italics, and the responses are in **blue regular font**. In each response, we refer to the subsections in the manuscript that have been adapted accordingly.

REVIEWER COMMENTS

Reviewer #2 (Remarks to the Author):

The authors have improved the manuscript significantly improved their manuscript, and addressed most of my concerns. However, I have a few remaining concerns.

1. Comparison with PointNovo. The authors said PointNovo does not provide the model parameters, and thus it is not easy for a straightforward comparison. They further claimed Casanovo performs better than PointNovo. However, from the current version of Casanovo manuscript, it is hard to tell the performance gaining in most of the tested cases as it only used the publish results when prec-recall of PointNovo, while the casanovo results were shown with different prec/recall. Ideally, the performance comparison should be compared using the prec/recall curve. In fact, Pointnovo authors have provided the training code and training dataset, which make it easy to reproduce the model parameters.

Response:

We agree with the reviewer that precision-recall curves allow a better performance comparison between Casanovo and PointNovo, instead of single precision and coverage values as presented in the Casanovo manuscript. Therefore, we have now downloaded PointNovo's code from the Zenodo repository (<https://zenodo.org/record/3960823>) and trained the model on the nine-species dataset with a leave-one-species-out cross-validation strategy to first reproduce the original models presented in their paper. **We have applied these models to the previously introduced test set of the human dataset by Wang et al. and to the nine-species dataset.** To avoid data leakage in the evaluation, we subset the nine-species dataset to contain only peptide-spectrum-matches for which the correct sequence is not present in the training set of the dataset by Wang et al., as suggested by reviewer #3. Furthermore, for the evaluation of each species, we removed peptides that were contained in any of the other species and also those spectra for which the experimental mass differed from the mass of the correct peptide by more than one Dalton. This is now described in the Methods section of our manuscript.

Altogether, we have now applied Spectralis-score to peptides proposed by each of the four de novo sequencing tools Novor, DeepNovo, Casanovo and PointNovo on both datasets (Wang et al. and nine-species dataset). Applying Spectralis-score on the nine-species dataset increases, though modestly, recall at high precision ranges and significantly improves the average precision (AUPRC). This improvement shows that Spectralis generalizes to different species, experimental configurations and de novo sequencing tools that were never considered during training. We adapted the results section "Levenshtein distance estimate improves PSM scoring" accordingly and appended the evaluation of Spectralis-score on PointNovo and DeepNovo on both datasets to the Supplementary Material of our manuscript (updated Supplementary Fig. 13, and new Supplementary Figs. 15-17).

In addition, to make it a fair comparison, it would also be useful to investigate if different training data used by different models will impact the performance of de novo sequencing algorithms.

Response:

While comparing the model architectures by retraining every model of different datasets would be interesting, we think that this is out of the scope of this work. Indeed, it is notoriously difficult to train deep learning models. Every new dataset requires investigating hyper-parameter optimization. Instead, we compare the models as they have been trained by the original authors. This matches the typical use cases of users of those tools. In this revised version of the manuscript, we have expanded the evaluation of our tool to peptide identifications by PointNovo on the datasets by Wang et al. and Tran et al. and to peptide identifications by DeepNovo on the dataset by Tran et al. As mentioned in

the previous point, we found that Spectralis-score also improves the ranking of candidate peptides of other tools on different datasets.

2. The paper shows that the rescoring PSMs using Spectralis-score improves the recall of de novo sequencing algorithms. I wonder if it is due to the re-rank of the candidate peptides sequenced on the same spectra, or the adjustment of the score distributions among the best PSMs of different spectra. If it is the former case, on what percentage of the cases, the improvement is due to swap of adjacent amino acid residues? If it is the latter case, how is the improvement correlated with the length of the sequenced peptides?

Response:

Improving the recall with Spectralis-score by re-ranking peptides sequenced on the same spectra ("the former case") does not apply because none of the methods we considered provide multiple candidates per spectra. Indeed, the tools Novor, DeepNovo, PointNovo and Casanovo only output at most one peptide per spectrum. The Casanovo-Novor combination also consists of a single candidate peptide spectrum, namely Casnovo if the mass matches the precursor and Novor otherwise. We have now clarified this in the section "Levenshtein distance estimate improves PSM scoring" of the manuscript.

We have now investigated the correlation between the improvement of rescoring the best PSMs of different spectra ("the latter case") for Casano candidates with peptide lengths (new Supplementary Fig. 8). This shows that improvements are much larger for longer peptides. This is expected as longer peptides are more difficult to sequence by de novo sequencing tools and, therefore, the tools make more mistakes on those. We added a sentence regarding this manner to the subsection "Levenshtein distance estimate improves PSM scoring"

3. Can the model to predict the Spectralis-score be extended to predict the score each positions in the sequenced peptides? It seems the features contains sufficient information for such prediction, while the positional score will give the indication of the accuracy of each sequenced residue.

Response:

This is an interesting idea but would likely require major new developments. While the bin classification model provides confidence scores in the m/z space, i.e. to the peaks, this does not easily map to individual amino acids in the sequence space. Unless for very specific exceptions (swap of adjacent amino acids), differences in individual amino acids lead to widespread shifts of m/z peak locations. Moreover, even a swap of adjacent amino acids can lead to drastic differences in peak intensities. Therefore, providing single amino acid level confidence scores is out of the scope of this study.

Reviewer #3 (Remarks to the Author, see also attached files):

The revised version of the manuscript was responsive to the reviewers' comments for the first submission, and the revised content significantly improves the presentation and characterization of the proposed approach. However, the improved characterization of Spectralis also revealed important new concerns that need to be addressed.

First, the new results for the 9 tissues datasets show a much lesser degree of improvement for Spectralis over Casanovo for a fair number of species, which brings into question the generalizability of the Spectralis models.

Response:

To facilitate the comparison, we are now introducing scatterplots to show the performance measurements before and after rescoring. Rescoring Casanovo v2 on the nine-species dataset with

Spectralis-score improves the average precision (AUPRC) eight of nine species (new Supplementary Fig. 16). Improvements are also for the recall at 90 % precision for seven of nine species (new Supplementary Fig. 17). Therefore, even though the degree of improvement on this data is less compared to the dataset by Wang et al., there is generally added value of Spectralis also on the nine-species dataset, despite being trained on the Wang et al. dataset in contrast to the other methods that were trained on the nine-species dataset. We modified the results section "Levenshtein distance estimate improves PSM scoring" in the manuscript accordingly.

Furthermore, it was not clear from the text whether the evaluation on the 9 tissues dataset removed all Peptide Spectrum Matches (PSMs) for all precursors (i.e., distinct pairs of modified sequence and charge state) which were included in the Spectralis training set due to sequence conservation between human and the other species. If these precursors were not excluded from the evaluation set then this should be revised to evaluate the performance of Spectralis without these precursors on the 9 species dataset. The revised version of the manuscript should include Spectralis predictions (as with the original submission for the PXD010154 dataset), as well as including columns for the spectrum files, scan numbers and ground truth identifications for all the PSMs used for the evaluation.

Response:

We thank the reviewer for raising this point. We have now removed PSMs for which the correct peptide sequences were included in the set of correct peptides of the training set of the human dataset by Wang et al. from the evaluation on the nine-species dataset. Moreover, for each species, we have removed all PSMs for which the corresponding correct sequence was included in any other species, as suggested in Casanovo's manuscript. **We updated Supplementary Figures 14 and 25 accordingly.** The improvement after rescoring with Spectralis-score on this modified nine-species dataset remained similar for all species but human and mouse, as somewhat expected due to sequence conservation between human and mouse. We adapted the section Methods and the results section "Levenshtein distance estimate improves PSM scoring" in the manuscript accordingly. Furthermore, we now added the evaluation files (containing spectrum identifiers, ground truth and predicted peptides, as well as original score and Spectralis-score) to the Zenodo repository. We remark that the new scatterplots, as well as the PointNovo and DeepNovo benchmarking (new Supplementary Figs. 15-17) were created based on this modified nine-species dataset.

Second, the concerns with generalizability led the reviewer to a closer inspection of the data submitted to Zenodo, which revealed two new substantial concerns when cross-referenced with the MaxQuant identifications submitted with the original PXD010154 dataset whose data was used for both training and evaluating the models presented in this manuscript.

The first concern is that the total number of PSMs identified in human heart tissue by Maxquant for 4,033 precursors is higher than the number of PSMs for the same precursors reported in the Spectralis predictions, which leaves serious concerns that the remaining PSMs may have somehow been included in the training datasets, which would have been a serious problem with the training/evaluation procedures (as the authors correctly acknowledged in the response to reviewer #2). As shown in the P013201_heart_precursors_reused.txt file provided with the review, there are many precursors with hundreds of PSMs in the same files in the original dataset for which there are also a subset of PSMs and predictions in the Spectralis set of reported results, which should not have been possible if the evaluation was performed as described in the manuscript and as replied to reviewer #2.

Response:

We thank the reviewer for the closer inspection of the dataset, which has helped us to improve and expand the description of our data processing procedure in the Methods section of the manuscript and to add important additional information to the data submitted to Zenodo. After

carefully comparing the reviewer's files (P013201_heart_precursors_reused.txt and P013201_heart_spectralis_vs_maxquant.txt) with our submission to Zenodo and the original MaxQuant file (obtained from the compressed file 9healthy_human_tissues_Ensembl_txt.zip in the PRIDE repository PXD010154) we conclude the following:

- We mapped a different MaxQuant peptide than the reviewer to 310 out of 36,312 PSMs in the tissue heart. We attribute this difference to the fact that in a few cases, MaxQuant proposes a secondary peptide to a spectrum (marked as "MULTI-SECPEP" in the column "Type"). In our work, and as originally described in the Methods section of our manuscript, we discarded these secondary peptides to ensure that at most one correct peptide sequence is associated with each spectrum. In the reviewer's file, some spectra are mapped to these secondary peptides. For instance, in the reviewer's file "P013201_heart_spectralis_vs_maxquant.txt", the spectrum from raw file 01296_D04_P013201_S00_N27_R1 and scan number 32635 is mapped to the peptide VGAHAGEYGAEALER which is marked as a secondary peptide by MaxQuant, while we consider the peptide FGMFEFLSNHMR as the only correct peptide for that spectrum. This different mapping of correct peptides leads to a difference in the computed precision and recall values, as well as the number of PSMs computed for each precursor. We improved the wording in the subsection "Data preprocessing" of the Methods section of our manuscript accordingly.
- Moreover, we realized that we had to be clearer and more precise on the filtering criteria that we applied to the data. We indeed applied additional filtering criteria to the data that we did not explicitly mention, which resulted in a smaller dataset compared to the original MaxQuant file. However, these additional filtering criteria do not raise any concerns regarding overfitting to the test set, since at all times we ensured that no MaxQuant peptide was present in both train and test set. We have now expanded the subsection "Data preprocessing" of the Methods section to include all filtering criteria.

The second concern is that a direct comparison of the Spectralis predictions submitted to Zenodo with the MaxQuant identifications submitted with the original PXD010154 dataset yields a much worse Precision/Recall curve than what is reported in the manuscript (e.g., Figure 5c). As shown in the P013201_heart_spectralis_vs_maxquant.txt file provided with the review, the highest Recall observed at Precision 90% is Recall 41.3%, which appears to be lower than the values depicted in Figure 5c.

Response:

We attribute the difference in the computed precision and recall values of Spectralis-EA to the different mapping of ground truth peptides to the spectra due to secondary peptides, as described above. **We have now appended the MaxQuant sequence to the data uploaded to Zenodo and additionally uploaded the precision and recall computation for the heart sample.** Here, consistent with the green curve in Fig. 5C, we compute 43% recall at 90% precision (instead of 41.3% as computed by the reviewer). We attach the difference in the computation of precision-recall curves by the reviewer and by us below, which as stated above, is attributed to the different mapping of MaxQuant sequences to the data. We do not see that the described difference in mapping leads to a much worse precision-recall curve. Nevertheless, if the reviewer found a larger difference somewhere else that we are overseeing, we would appreciate the feedback.

Furthermore, we uploaded Spectralis-score on the combination of Novor and Casanovo sequences (Casanovo-Novor, violet line in Fig. 5C) to Zenodo in order to allow comparison with the scores computed on the sequences obtained from Spectralis-EA (green curve in Fig. 5C).

The most likely explanation for both of these concerns is that the authors likely used a different set of MaxQuant identifications than what was reported in the original dataset submission, possibly by refining those identifications with additional filters or criteria that were not clear in the manuscript. The simplest way to address these concerns would be to explain the differences by adding the following to the Zenodo data: (a) submit the full set of MaxQuant identifications used for training the models and (b) add columns to the Spectralis results submitted to Zenodo to also show the ground truth MaxQuant identifications that are being used to evaluate the performance of the models.

Response:

We thank the reviewer for their suggestion. As described above, we indeed used a different set of MaxQuant identifications than the ones used by the reviewer. **We improved the wording regarding the secondary peptides and described all filtering criteria that were applied to the data. Moreover, we expanded our Zenodo submission. We believe that this will allow more clarity and transparency regarding our model training and evaluation.** The Zenodo repository (with DOI [zenodo.8393846](https://doi.org/10.5281/zenodo.8393846)) now contains the following files:

- For the human dataset by Wang et al.:
 - **train_val_test_split.csv** containing the mapping of the correct peptide by MaxQuant to either train, validation or test set
 - **psms_train_val_test.csv** containing the mapping of correct PSMs (scan number, raw file and correct peptide by MaxQuant) to either train, validation or test set
 - **updated_spectralis_test_out.csv** as before containing Spectralis-EA predictions and scores on test set, as well as initial peptides and scores by Casanovo and Novor and now containing also correct peptides by MaxQuant and Spectralis-scores on the combination of Casanovo and Novor sequences (column named `spectralis_score_onlyRescoring`)
 - **spectralis_test_out_heart_analysis.csv** subset of `20220822_spectralis_test_out.csv` containing only PSMs for the tissue heart with the computation of precision and recall values
 - **spectralis_test_out_pointnovo_deepnovo.csv** containing predictions by DeepNovo and PointNovo with original scores and Spectralis-score, as well as correct peptides by MaxQuant
- For the nine-species dataset by Tran et al.:
 - **spectralis_ninespecies_out.csv** containing spectrum identifiers, correct peptides by PEAKSDB, predicted peptides by the different de novo sequencing tools as well as original scores and Spectralis-scores for the different PSMs.

In all files, Leucine residues in the correct and predicted sequences are replaced by Isoleucine.

Finally, the evaluation of the data submitted with the manuscript also prompted an attempted evaluation of the code submitted with the manuscript, which actually revealed that the code does not work and is not usable even by developers, let alone by biologists or mass spectrometrists attempting to use the proposed approach. While the given link to the author's GitHub repository is accessible, there was no detailed description about the tool inside the repository or in the manuscript. More detailed inspection of the code (which is also almost completely undocumented) further revealed that the shared code relies on the Prosit spectrum predictor which is accessed through remote procedure calls using certificate files. As such, even if the code worked (which it did not), it appears like users

would not be able to use Spectralis as a standalone tool, and may not even be able to use it free of charge for their own data. While this is not necessarily a reason to prevent publication of the proposed approach, it does significantly reduce the potential impact of the proposed approach.

Response:

We thank the reviewer for raising this, which has made us substantially improve the code repository. We have now included a README.md file to the public GitHub repository which includes a **brief description of the tool, installation instructions and examples on how to use the code.** Furthermore, Spectralis is now easily installable with pip and runnable via Python code or command line interface. We also provided an example dataset to test the tool in the GitHub repository (example.mgf). **Importantly, we have modified and tested the code so that it no longer depends on certificate files**, but instead uses Koina (<https://koina.proteomicsdb.org/>) which easily and freely allows the collection of Prosit predictions. We believe that this will make Spectralis easily usable and runnable by others.

Reviewers' Comments:

Reviewer #2:

Remarks to the Author:

The authors have addressed my concerns adequately. I do not have further comments.

Reviewer #3:

Remarks to the Author:

The authors have satisfactorily addressed all concerns - the revised manuscript provides an accurate description of the methods and results and is suitable for publication as is.

Response to reviewers for “NCOMMS-23-02312-B”

REVIEWER COMMENTS

Reviewer #2 (Remarks to the Author):

The authors have addressed my concerns adequately. I do not have further comments.

We once again thank the reviewer for their constructive feedback and careful inspection of our manuscript.

Reviewer #3 (Remarks to the Author):

The authors have satisfactorily addressed all concerns - the revised manuscript provides an accurate description of the methods and results and is suitable for publication as is.

We once again thank the reviewer for their constructive feedback and careful inspection of our manuscript.